# TexTailor: Customized Text-aligned Texturing via Effective Resampling

**Suin Lee, Dae-Shik Kim**
KAIST
Daejeon, South Korea
{suinlee, daeshik}@kaist.ac.kr

## Abstract

We present *TexTailor*, a novel method for generating consistent object textures from textual descriptions. Existing text-to-texture synthesis approaches utilize depth-aware diffusion models to progressively generate images and synthesize textures across predefined multiple viewpoints. However, these approaches lead to a gradual shift in texture properties across viewpoints due to (1) insufficient integration of previously synthesized textures at each viewpoint during the diffusion process and (2) the autoregressive nature of the texture synthesis process. Moreover, the predefined selection of camera positions, which does not account for the object's geometry, limits the effective use of texture information synthesized from different viewpoints, ultimately degrading overall texture consistency. In TexTailor, we address these issues by (1) applying a resampling scheme that repeatedly integrates information from previously synthesized textures within the diffusion process, and (2) fine-tuning a depth-aware diffusion model on these resampled textures. During this process, we observed that using only a few training images restricts the model's original ability to generate high-fidelity images aligned with the conditioning, and therefore propose an performance preservation loss to mitigate this issue. Additionally, we improve the synthesis of view-consistent textures by adaptively adjusting camera positions based on the object's geometry. Experiments on a subset of the Objaverse dataset and the ShapeNet car dataset demonstrate that TexTailor outperforms state-of-the-art methods in synthesizing view-consistent textures. The source code for TexTailor is available at https://github.com/Adios42/Textailor

## 1 Introduction

Realistic, high-quality 3D creatures are critical for creating immersive experiences in video games, films, and AR/VR applications, making them an essential part of modern digital media. While advancements in graphics engines and technical expertise have enabled the production of high-quality 3D content, the process remains labor-intensive, requiring multiple iterations and adjustments as well as substantial creative input.

To alleviate these challenges, the computer vision community has focused on breakthroughs in implicit neural representations (Mildenhall et al., 2021; Barron et al., 2021; Chen et al., 2022; Wang et al., 2021) and diffusion models based on textual descriptions, which provide an intuitive approach to 3D content generation (Rombach et al., 2022; Saharia et al., 2022; Nichol et al., 2021). Notably, the introduction of the score distillation sampling (SDS) loss function (Poole et al., 2022) has enabled the generation of diverse, high-quality 3D content by combining implicit neural representations with the strong priors provided by diffusion models (Wang et al., 2024; Lin et al., 2023).

While these methodologies provide both geometry and texture, converting implicit neural representations into explicit formats, such as meshes, remains necessary for integration into graphics engines and real-time applications. Recently, DMTet (Shen et al., 2021) has enabled precise mesh geometry extraction from implicit representations by leveraging a signed distance field and the Marching Tetrahedra algorithm. However, in texture synthesis, texture unwrapping often leads to inconsistent mappings, which can degrade the visual quality of the output or necessitate additional texture synthesis steps (Lin et al., 2023; Chen et al., 2023b).

With significant advances in 3D geometry generation (Shen et al., 2021; Vahdat et al., 2022; Chen & Zhang, 2019; Nash et al., 2020; Müller et al., 2023) and geometry optimization process (Shen et al.,

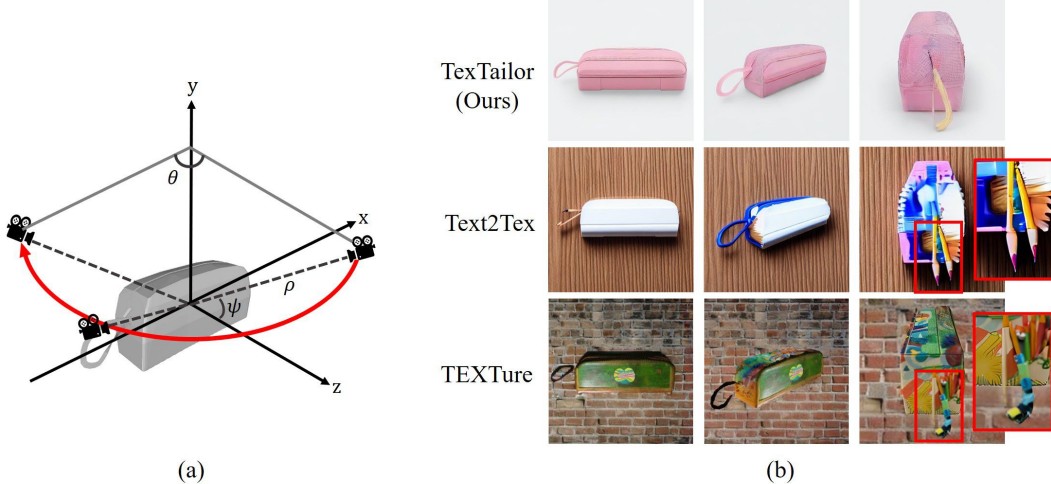

(a)             (b)

Figure 1: **(a)**: The illustration of definition of viewpoint. Following the red arrow, the pencil case mesh is painted from left to right in each row in (b). **(b)**: This visualization illustrates the gradual shift in texture properties that becomes more pronounced as the viewpoint changes. Compared to Text2Tex (Chen et al., 2023a) and TEXTure (Richardson et al., 2023), TexTailor exhibits significantly less of this gradual shift.

2021), recent research (Chen et al., 2023a; Richardson et al., 2023; Tang et al., 2024; Youwang et al., 2024; Metzer et al., 2023) has focused on texture synthesis strategies for textureless meshes using language cues. Among these approaches, several works (Tang et al., 2024; Chen et al., 2023a; Richardson et al., 2023) utilize inpainting techniques (Lugmayr et al., 2022) within pre-trained depth-aware image diffusion models (Zhang et al., 2023; Rombach et al., 2022) to progressively generate images, projecting them back onto the mesh for specific regions from predefined viewpoints. However, achieving coherent texture synthesis across all viewpoints remains a challenge for two key reasons: (1) The current inpainting techniques applied to texture synthesis are incomplete, as information from previously synthesized visible textures at the current viewpoint is reflected onto the untextured areas only once per timestep in the diffusion process. This results in inconsistencies between adjacent viewpoints. (2) Additionally, progressively generating textures from multiple viewpoints introduces inherent sequential and temporal biases, gradually obscuring the texture information from the initial viewpoint. As a result, the characteristics of the texture synthesized from the initial viewpoint gradually degrade as the viewpoints shift, a phenomenon similar to the short-term dependency problems (Sutskever, 2014) observed in language models.

For instance, assuming the Y-axis is the vertical axis in the world coordinate system, we define the viewpoint as $v = (\theta, \psi, \rho)$, as shown in Fig.1 (a). Fig.1 (b) shows the generated texture images of 'a pencil box' mesh as $\theta$ decreases, with $\psi$ and $\rho$ fixed, following red arrow in Fig. 1 (a). In comparison to **TexTailor** in Fig. 1(b), the texture properties (e.g., color and pattern) of the other methods, synthesized from the initial viewpoint, progressively degrade as $\theta$ decreases. This gradual degradation intensifies over time, ultimately undermining texture consistency.

To address these challenges, we present a novel texture synthesis method, **TexTailor**. (1) **TexTailor** utilizes the resampling scheme proposed by Lugmayr et al. (2022) for texture synthesize fields, which helps to repeatedly integrate information from previously synthesized textures. To enhance its effectiveness, we implement this scheme within the non-Markovian diffusion process (DDIM, Song et al. (2020)). And (2) we finetunes a depth-aware text-to-image diffusion model using images generated through this resampling. During this process, we observe that finetuning with a limited sample set risks reducing the model's ability to generate high-fidelity images conditioned on depth maps and textual descriptions. To mitigate this, we introduce an *performance preservation loss*, ensuring that the model learns the resampled distributions without compromising its generalization capabilities. Additionally, since the inpainting technique relies on previously synthesized visible textures, poorly predefined viewpoint locations that fail to consider the complexity of the mesh geometry can lead to suboptimal texture generation for the current viewpoint. To resolve this, we propose an *adaptive viewpoint refinement* scheme, which dynamically adjusts the viewpoint based on the amount of previously generated texture. This method enables the synthesis of coherent textures across all viewpoints by adaptively positioning the viewpoint according to the geometry of the mesh.

Experiments on a subset of the Objaverse dataset (Deitke et al., 2022) and the ShapeNet car dataset (Chang et al., 2015) demonstrate the superior performance of **TexTailor** in synthesizing coherent 3D textures guided by textual descriptions. It surpasses state-of-the-art texture synthesis methods driven by language cues, achieving better results in terms of LPIPS (Zhang et al., 2018) and FID (Heusel et al., 2017) on a subset of the Objaverse dataset. Our contributions are as follows:

- We extend the resampling technique, previously used only in 2D image inpainting, to the field of 3D texture synthesis by applying it to the DDIM non-Markovian process, enabling consistent texture generation within a single view with significantly fewer steps.

- We analyze the gradual shift phenomenon in the texture synthesis process and propose a novel approach that trains the model using only a few resampled images, removing the need for an external dataset of 3D meshes, textures, or text. By incorporating the performance preservation loss, this method effectively mitigates the gradual shift across multiple angles.

- To address the catastrophic forgetting phenomenon, we propose the performance preservation loss to mitigate this issue and maintain the model's performance. Experimental results demonstrate that this loss effectively prevents forgetting and maintains high texture quality across diverse viewpoints.

- We introduce an adaptive method that dynamically adjusts camera positions based on texture coverage, eliminating manual configuration and ensuring consistent texture synthesis for complex geometries.

## 2 BACKGROUND

### 2.1 DIFFUSION MODELS

Diffusion models generate high-fidelity images by learning to iteratively convert a sample from a simple Gaussian distribution $\boldsymbol{x}_T \sim \mathcal{N}(\mathbf{0}, \boldsymbol{I})$ into a complex data distribution $\boldsymbol{x}_0 \sim q(\boldsymbol{x}_0)$ through a forward(diffusion) and generative process (Ho et al., 2020; Sohl-Dickstein et al., 2015; Song et al., 2020). The forward process, which approximates the posterior $q(\boldsymbol{x}_{1:T}|\boldsymbol{x}_0)$, is modeled as a Markov chain that progressively adds Gaussian noise to the data using coefficients $\bar{\alpha}_{1:T} \in (0, 1]^T$:

$$q(\boldsymbol{x}_{1:T}|\boldsymbol{x}_0) := \prod_{t=1}^{T} q(\boldsymbol{x}_t|\boldsymbol{x}_{t-1}), \text{where } q(\boldsymbol{x}_t|\boldsymbol{x}_{t-1}) := \mathcal{N}\left(\sqrt{\frac{\bar{\alpha}_t}{\bar{\alpha}_{t-1}}}\boldsymbol{x}_{t-1}, \left(1 - \frac{\bar{\alpha}_t}{\bar{\alpha}_{t-1}}\right)\boldsymbol{I}\right). \quad (1)$$

The generative process, modeled by the joint distribution $p_\theta(\boldsymbol{x}_{0:T})$, gradually denoises noisy data, starting from $p(\boldsymbol{x}_T) := \mathcal{N}(\boldsymbol{x}_T; \mathbf{0}, \boldsymbol{I})$. During this process, a noise predictor $\epsilon_\phi(\boldsymbol{x}_t; t)$, typically implemented using a U-Net architecture (Ronneberger et al., 2015), predicts the noise in $\boldsymbol{x}_t$. Song et al. (2020) introduce DDIM, a non-Markovian diffusion process that enables different generative samplers by adjusting the generative noise variance, allowing for deterministic mappings. This approach reduces the number of sampling steps while maintaining the same marginals as DDPM.

However, training diffusion models in high-dimensional pixel space can be computationally expensive. To address this, Rombach et al. (2022) proposed Stable Diffusion Model, which first use auto-encoding models to transform the data into a lower-dimensional semantic space before applying the forward and generative processes. This dimensionality reduction significantly decreases the computational cost. Consequently, the noise predictor is trained in the semantic space as follows:

$$L_{LDM}(\phi, \boldsymbol{z}_0) := \mathbb{E}_{\boldsymbol{z}_0, \epsilon \sim \mathcal{N}(\mathbf{0}, \boldsymbol{I}), t \sim \mathcal{U}(0,1)}\left[w(t)\|\epsilon_\phi(\sqrt{\bar{\alpha}_t}\boldsymbol{z}_0 + \sqrt{1 - \bar{\alpha}_t}\epsilon; t, \boldsymbol{c}) - \epsilon\|_2^2\right], \quad (2)$$

where $\boldsymbol{z}_0$ is the latent vector of the input image $\boldsymbol{x}_0$, obtained from the auto-encoder, $\boldsymbol{c}$ is a conditioning vector (e.g., from textual descriptions), and $w(t)$ is a weighting term indexed by timestep $t$. Zhang et al. (2023) extends this approach by adding trainable encoder blocks connected to convolutional layers with zero weights (ControlNet) to the pre-trained noise predictor network $\epsilon_\phi(\boldsymbol{z}_t, t, \boldsymbol{c})$. This architecture ensures that the pre-trained diffusion model remains fixed during training, while the newly added trainable blocks and layers learn conditional information, such as depth, normal, and edge maps. This approach enables the model to generate data distributions controlled by additional 2D spatial inputs including depth maps, allowing training with fewer parameters on smaller datasets without compromising pre-trained capabilities. In this paper, we utilize Stable Diffusion with ControlNet as a depth-aware T2I diffusion model, fine-tuning only the ControlNet component to synthesize partial textures conditioned on depth maps.

## 2.2 TEXT-TO-TEXTURE GENERATION

**Texture synthesis procedure.** In recent works (Richardson et al., 2023; Chen et al., 2023a; Tang et al., 2024), the partial mesh surface viewed from a single viewpoint are segmented into several regions and undergo incremental texturing from one viewpoint to the next to ensure both local and global consistency. Specifically, Chen et al. (2023a) divides the partial surface into four regions ("keep", "new", "update", and "ignore") as illustrated in Fig. 2 (b), and synthesizes partial texture using an image inpainting technique. The portion of the surface in the current viewpoint

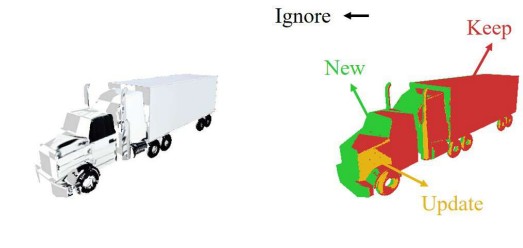

(a) Current view point image     (b) Four regions

Figure 2: **Left**: The image $x_0$ rendered from the current viewpoint before being processed by the depth-aware diffusion model. **Right**: An illustration of the four regions on the partial mesh surface.

that has already been textured from the previous viewpoint(see Fig. 2(a)) is labeled the "keep" region, while the part that lacks texture is called the "new" region. Only the "new" region undergoes partial texturing using the inpainting strategy, while the "keep" region remains unchanged. Additionally, from the same viewpoint, surfaces where view direction is parallel to the normal vector of the object's visible faces are categorized as the "update" region. This "update" region is re-textured and projected again since it is viewed from a better angle than other regions. Consequently, partial mesh surfaces corresponding to the "new" region are textured using information from the "keep" region across multiple viewpoints.

**Image inpainting for texturing.** During the texture synthesize process, an image inpainting strategy (Lugmayr et al., 2022) is applied to a depth-aware diffusion model (Rombach et al., 2022; Zhang et al., 2023) to effectively texture the missing regions of the mesh surface corresponding to the "new" region:

$$z_{t-1}^{\text{known}} \sim \mathcal{N}(\sqrt{\bar{\alpha}_t}z_0, (1 - \bar{\alpha}_t)\boldsymbol{I}), \tag{3}$$

$$z_{t-1}^{\text{unknown}} \sim \mathcal{N}(\mu_\phi(z_t, t), \sigma_\phi(z_t, t)), \tag{4}$$

$$\tilde{z}_{t-1} := z_{t-1}^{\text{known}} \odot (1 - \mathcal{M}_{latent}) + z_{t-1}^{\text{unknown}} \odot \mathcal{M}_{latent}, \tag{5}$$

where $z_0$ is the latent vector of the input image $x_0$, which is rendered from the current viewpoint of an object with texture applied from a previous viewpoint (see Fig. 2(a)). $z_{t-1}^{\text{known}}$ is the latent vector sampled at timestep $t - 1$ by adding noise to $z_0$, while $z_{t-1}^{\text{unknown}}$ is the model's output at the same timestep. $\mathcal{M}_{latent}$ represents a mask, with a value of 1 for parts corresponding to the unknown region("new" $\cup$ "ignore") on the image plane(see Fig. 2), resized to match the size of the latent space. In the latent space, by combining $z_{t-1}^{\text{unknown}}$ for the unknown region and $z_{t-1}^{\text{known}}$ for the known region("keep"), the result is the composite $\tilde{z}_{t-1}$.

This process generates a partial texture for the current viewpoint based on information from previous viewpoints. However, simply merging the known and unknown regions once at each timestep is insufficient to fully harmonize the texture information synthesized from previous viewpoints during the denoising process of the diffusion model.

## 3 METHOD

### 3.1 RESAMPLING

As mentioned in Sec. 2, texturing an object involves progressively generating images using a depth-aware diffusion model and compositing them across multiple viewpoints. During the denoising process at each timestep, simply merging the known and unknown regions once is insufficient to seamlessly harmonize the unknown regions with the known ones, leading to inconsistencies between adjacent viewpoints. Additionally, discordant textures synthesized from a single viewpoint can accumulate through subsequent viewpoints, further degrading overall 3D texture consistency.

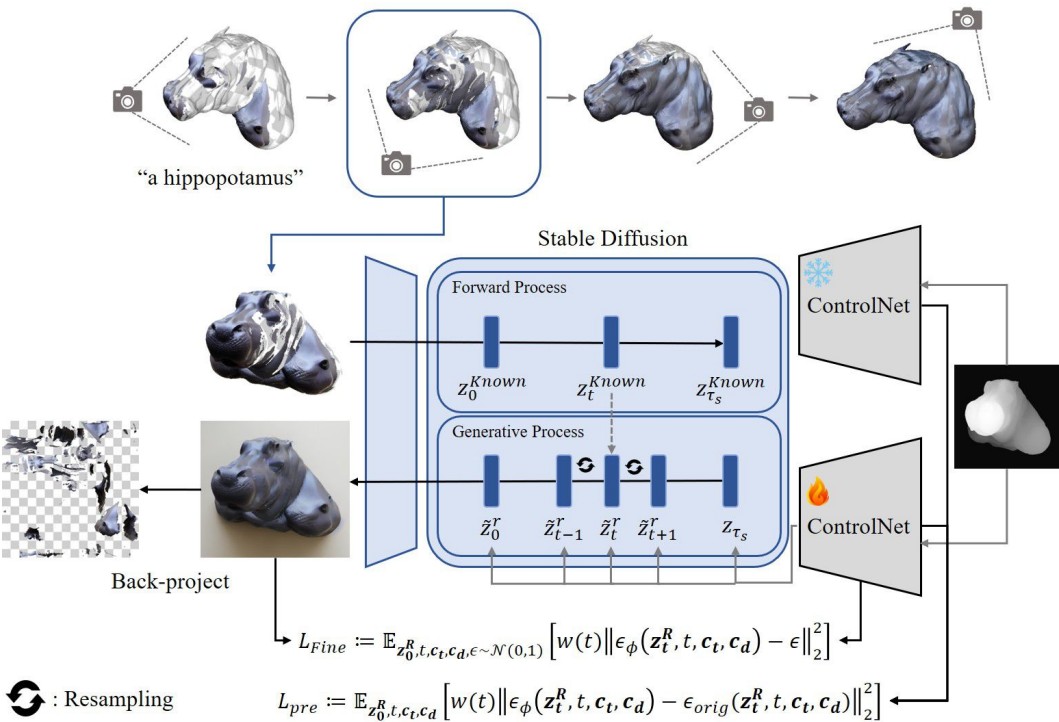

Figure 3: **Overview of TexTailor.** TexTailor synthesizes textures for a given 3D mesh without textures, based on a textual description, such as "a hippopotamus". We add additional camera positions to the predefined set to properly condition the previously synthesized textures from each viewpoint, eliminating the need for manually coordinating camera locations. Based on these viewpoints, we progressively generate textures using image inpainting techniques, including a resampling scheme within a non-Markovian process. To prevent the gradual shift in texture properties, we fine-tune ControlNet with a small set of resampled images, incorporating an performance preservation loss.

In 2D image inpainting field, Lugmayr et al. (2022) propose a resampling method that iteratively repeats adding noise and denoising, along with a merging step (similar to Eq. 5), multiple times at each timestep to more thoroughly mix the known and unknown regions. However, directly applying this method to texture synthesis requires a large number of timesteps, as it operates within a Markovian process, similar to DDPM, and must be repeated across multiple viewpoints. To generate higher quality, harmonious images in fewer steps, we propose a resampling method within a non-Markovian process using the following formula:

$$\tilde{z}_t^r \sim \mathcal{N}\left(\sqrt{\frac{\bar{\alpha}_t}{\bar{\alpha}_{t-1}}}\tilde{z}_{t-1}^{r-1}, (1 - \frac{\bar{\alpha}_t}{\bar{\alpha}_{t-1}})\boldsymbol{I}\right), \tag{6}$$

$$\tilde{z}_{t-1}^r \sim \mathcal{N}\left(\sqrt{\bar{\alpha}_{t-1}}\tilde{z}_0^r + \sqrt{1 - \bar{\alpha}_{t-1}} \cdot \frac{\tilde{z}_t^r - \sqrt{\bar{\alpha}_t}\tilde{z}_0^r}{\sqrt{1 - \bar{\alpha}_t}}, 0\right), \tag{7}$$

$$\tilde{z}_{t-1}^r = z_{t-1}^{\text{known}} \odot (1 - \mathcal{M}_{latent}) + \tilde{z}_{t-1}^r \odot \mathcal{M}_{latent}, \tag{8}$$

where $r$ denotes the resampling step at each timestep, $r \in \{1, ..., R\}$, and $\tilde{z}_{t-1}^0$ corresponds to $\tilde{z}_{t-1}$ from Eq. 5. The resampling process at each timestep, as described in the equations above, occurs after the merging process defined by Eq. 5. Specifically, in the $r$-th resampling, $\tilde{z}_t^r$ is sampled by adding noise to $\tilde{z}_{t-1}^{r-1}$, the previously merged sample at timestep $t-1$, as shown in Eq. 6. In Eq. 7, $\tilde{\mathbf{z}}_{t-1}^r$ is sampled by denoising $\tilde{z}_t^r$ according to the DDIM non-Markovian process. The resulting $\tilde{\mathbf{z}}_{t-1}^r$ is then combined with $\mathbf{z}_{t-1}^{\text{known}}$, as described in Eq. 8, where $\mathbf{z}_{t-1}^{\text{known}}$ is derived from Eq. 3. The timestep $t$ is part of the sequence $\{\tau_s, ..., \tau_1\}$, which is a sub-sequence of $\{T, ..., 1\}$. This approach allows high-quality texture synthesis by merging the "known" regions and "unknown" regions $R$ times with only 30 steps. This is significantly fewer than the 250 steps required by the original resampling method (Lugmayr et al., 2022) for a single view.

## 3.2 Fine-tuning with Original Capabilities Preserved

Using the resampling method (Sec. 3.1) within the depth-aware diffusion model, the partial mesh surface from the current viewpoint are coherently painted based solely on the previously synthesized textures visible from that same viewpoint(corresponding to "keep" region). However, the autoregressive nature of the texture synthesis process, which relies only on previously synthesized visible textures, leads to a gradual shift in texture properties, such as material or pattern (see Fig.1(b)), ultimately compromising the consistency of the texture. This is because, as the synthesis progresses from the first viewpoint across multiple viewpoints, the texture information from the first viewpoint becomes increasingly obscured.

Furthermore, ControlNet (Zhang et al., 2023) is responsible for extracting data distributions that align with depth maps in the output domain of the diffusion model. However, it is challenging to guide resampled data distributions from the diffusion model through ControlNet without incorporating previously synthesized visible textures (i.e., without using image inpainting techniques) in the current viewpoint. In other words,

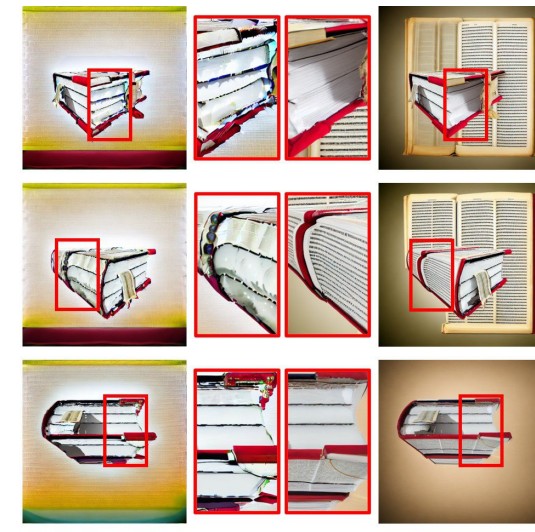

W/o preservation        TexTailor

Figure 4: **Illustration of the loss of original generative ability** when fine-tuning ControlNet without the performance preservation loss.

ControlNet itself does not retain or reflect the characteristics of previously synthesized resampled textures at the current viewpoint.

These limitations hinder the ability to infer texture information for viewpoints distant from the first viewpoint using resampled textures from viewpoints close to the first viewpoint. To address this challenge, we fine-tune ControlNet with a small set of resampled images near the first viewpoint to extract images of the same object from different angles in the output domain of the diffusion model, allowing it to retain the texture properties across multiple viewpoints:

$$L_{Fine} := \mathbb{E}_{\boldsymbol{z}_0^R, \epsilon \sim \mathcal{N}(\boldsymbol{0}, \boldsymbol{I}), t \sim \mathcal{U}(0,1), \boldsymbol{c}_t, \boldsymbol{c}_d} \left[ w(t) \| \epsilon_\phi(\boldsymbol{z}_t^R, t, \boldsymbol{c}_t, \boldsymbol{c}_d) - \epsilon \|_2^2 \right], \qquad (9)$$

where $\boldsymbol{c}_t$ and $\boldsymbol{c}_d$ are the text prompts and depth condition, respectively.

While this approach helps preserve texture properties during synthesis, it compromises ControlNet's original capability to extract images aligned with depth conditions and text prompts, a phenomenon known as catastrophic forgetting. For instance, the generated textures from the depth-aware diffusion model finetuned with Eq.9 exhibit significant noise and reduced quality, as shown in Fig.4. To address this issue, we introduce an performance preservation loss:

$$L_{pre} := \mathbb{E}_{\boldsymbol{z}_0^R, \epsilon \sim \mathcal{N}(\boldsymbol{0}, \boldsymbol{I}), t \sim \mathcal{U}(0,1), \boldsymbol{c}_t, \boldsymbol{c}_d} \left[ w(t) \| \epsilon_\phi(\boldsymbol{z}_t^R, t, \boldsymbol{c}_t, \boldsymbol{c}_d) - \epsilon_{orig}(\boldsymbol{z}_t^R, t, \boldsymbol{c}_t, \boldsymbol{c}_d) \|_2^2 \right], \qquad (10)$$

where $\epsilon_{orig}$ is the output of the fixed pre-trained network. This loss ensures that the noise prediction network does not deviate significantly from the original pre-trained parameters.

Consequently, Our final loss is as follows:

$$L_{Final} := L_{Fine} + \lambda L_{pre}, \qquad (11)$$

where $\lambda$ adjusts the strength of the performance preservation loss. Fig. 3 illustrates the entire process.

## 3.3 Adaptive Viewpoint Refinement

To achieve consistent texture synthesis using the inpainting method (Sec. 2.2), it is crucial to set the camera's position for each viewpoint appropriately. This is because the outcomes of images generated by the inpainting technique are highly dependent on the previously synthesized visible

textures at the current viewpoint. A simple solution is to distribute as many viewpoints as possible evenly around the object in 3D space. However, this approach presents two problems.

First, unnecessary predefined viewpoints make the overall texture synthesis process inefficient. Distributing too many viewpoints is akin to sequentially painting the entire mesh surface in small, redundant sections, which adds unnecessary steps and even exacerbates the gradual shift in texture properties, even with the introduction of Eq.11. Second, complex geometries can lead to camera positions where textures from previous viewpoints are not sufficiently visible. Finding evenly distributed viewpoints $\boldsymbol{v} = (\theta, \psi, \rho)$ in 3D space often requires numerous trials and errors to achieve precise viewpoint settings. Furthermore, optimal camera positions vary depending on the object's geometry, making manual configuration particularly difficult, especially for large-scale objects.

To select optimal camera positions, we utilize the ratio of the "keep" region to the "new" region at each viewpoint. We render each region from the meshes onto the image planes and convert them into masks where a value of one represents the corresponding region. Then, we calculate the proportion of pixels corresponding to the "keep" region relative to the total of the "keep" and "new" regions on the image plane:

$$p := \frac{\Sigma_{i,j}\mathbf{1}_{\{(i,j)\in\text{keep}\}}}{\Sigma_{i,j}\mathbf{1}_{\{(i,j)\in\text{new}\}} + \Sigma_{i,j}\mathbf{1}_{\{(i,j)\in\text{keep}\}}}. \tag{12}$$

If the proportion $p$ calculated in Eq. 12 is smaller than a threshold $\beta$, we add an additional camera position($\boldsymbol{v} = (\theta_{prev}+\gamma(\theta_{current}-\theta_{prev}), \psi_{prev}+\gamma(\psi_{current}-\psi_{prev}), \rho)$) to the predefined camera position sequence $\{\boldsymbol{v}_1, ..., \boldsymbol{v}_n\}$. Here, $\gamma \in (0,1)$ is an interpolation parameter, and we interpolate the azimuth ($\theta$) and elevation ($\psi$) angles between the previous and current viewpoints. This adaptive approach ensures that the camera positions are optimized based on the object's geometry and the amount of previously generated texture.

## 4 EXPERIMENTS

### 4.1 EXPERIMENT SETUP

**Implementation Details.** We select a subset of the Objaverse dataset (Deitke et al., 2022) to evaluate our model, following the approach of Chen et al. (2023a). In this dataset, Chen et al. (2023a) filter out low-quality or misaligned meshes from the designated categories, resulting in 410 textured meshes across 225 categories for our experiments. Notably, the original textures are used exclusively for evaluation. Recent text-driven texture synthesis methods (Richardson et al., 2023; Metzer et al., 2023; Youwang et al., 2024), which rely on gradient-based rendering using a differential renderer (Fuji Tsang et al., 2022), encounter difficulties when optimizing texture synthesis for 'car' objects from the ShapeNet dataset (Chang et al., 2015). Therefore, we present only qualitative results for our approach on this dataset to demonstrate its performance on fine-grained categories compared to Text2Tex.

For both datasets, the number of resampling steps, as well as the parameters $\lambda$, $\beta$, and $\gamma$, are set to 3, 2.5, 0.5, and 0.5, respectively. We finetune the ControlNet using five images rendered from viewpoints close to and including the first viewpoint: $\boldsymbol{v}_1 = (0°, 15°, 1)$, $\boldsymbol{v}_2 = (0°, 35°, 1)$, $\boldsymbol{v}_3 = (0°, -5°, 1)$, $\boldsymbol{v}_4 = (20°, 15°, 1)$, and $\boldsymbol{v}_5 = (340°, 15°, 1)$. For rendering and texture projection, we utilize the PyTorch framework (Paszke et al., 2017) along with PyTorch3D (Ravi et al., 2020).

**Evaluation metrics.** To quantitatively measure the view-consistency of synthesized textures, we report the average LPIPS (Zhang et al., 2018) between images rendered from 3D textured meshes across multiple viewpoints. A gradual shift in texture properties or the presence of artifacts is assumed to increase the perceptual loss between any two viewpoints. To calculate average LPIPS, We sample 25 uniformly spaced camera positions from both the upper and lower hemispheres of a fixed-radius sphere. All cameras within the upper hemisphere share the same elevation, and similarly, all cameras within the lower hemisphere share the same elevation, with each set directed towards the center of the sphere. An image is rendered from each viewpoint at a resolution of $512 \times 512$. We then compute the average LPIPS values for all pairs of images in the 3D scene and sum the averages across all evaluated categories.

Additionally, we assess the quality and diversity of the generated textures using the Frechet Inception Distance (FID) (Heusel et al., 2017). The synthesized texture distribution consists of rendered

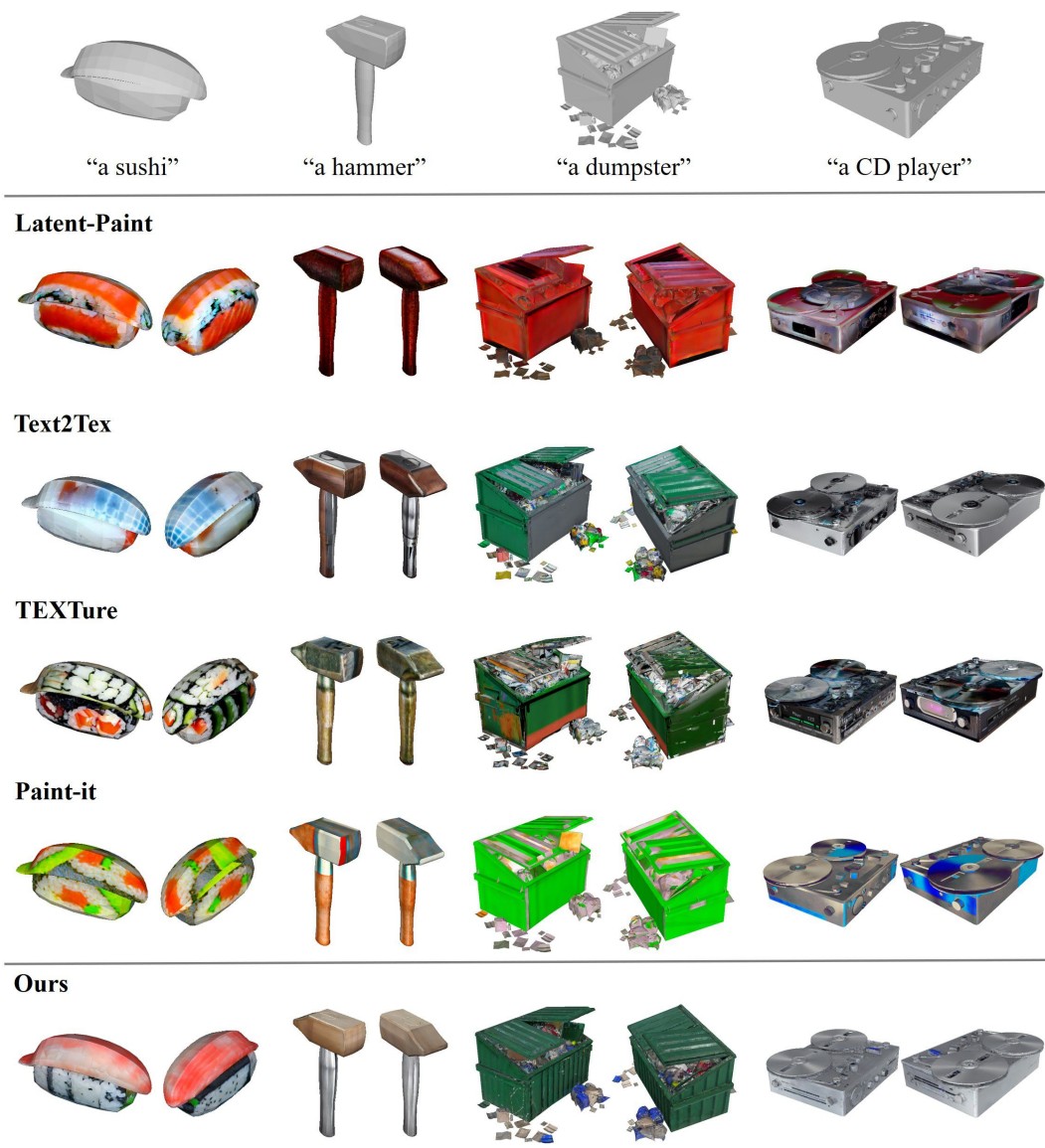

Figure 5: **Qualitative comparisons on Objaverse.** We compare TexTailor with state-of-the-art baselines (Metzer et al., 2023; Chen et al., 2023a; Richardson et al., 2023; Youwang et al., 2024) on Objaverse meshes. Compared to other methods, TexTailor produces textures that are more view-consistent and better aligned with object geometries.

images from the above camera points, while the real distribution is composed of renders of the meshes under the same conditions, using artist-designed textures.

**Quantitative comparisons.** In Tab. 1, we compare our approach against recent SOTA text-driven texture synthesis methods: Latent-Paint (Metzer et al., 2023), Text2Tex (Chen et al., 2023a), TEXTure (Richardson et al., 2023), and Paint-it (Youwang et al., 2024), on a subset of the Objaverse dataset. For texture descriptions, template, "*a ⟨category⟩*", is utilized consistently across all approaches. The results demonstrate that our method synthesizes high-quality and more consistent textures than all baselines across various categories.

Table 1: Quantitative comparison on Objaverse subset.

| Method | LPIPS↓ | FID↓ |
|---|---|---|
| Latent-Paint | 54.429 | 45.334 |
| Text2Tex | 38.931 | 33.487 |
| TEXTure | 54.138 | 43.337 |
| Paint-it | 38.316 | 39.823 |
| **Ours** | **37.889** | **29.998** |

**Qualitative comparisons.** In Fig. 5, we present the rendered results using **TexTailor** alongside other baselines on the Objaverse dataset. The qualitative results demonstrate that our approach significantly improves texture quality, particularly in terms of

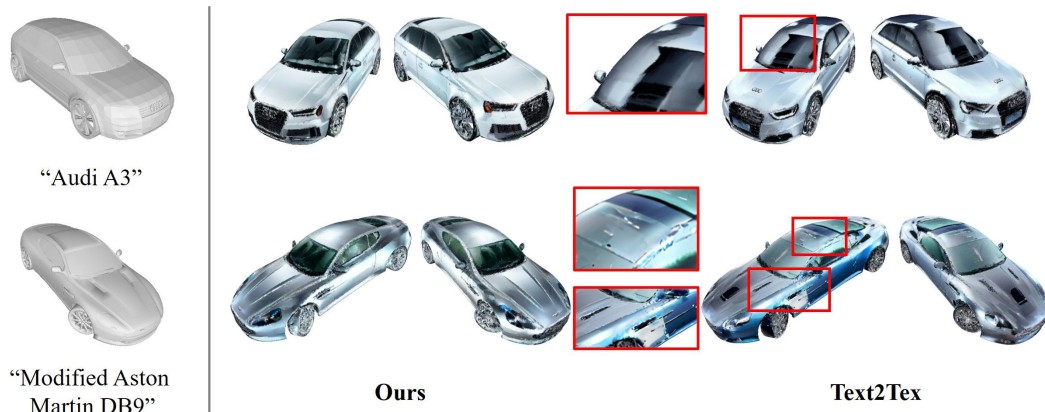

"Audi A3"

"Modified Aston Martin DB9"

**Ours**

**Text2Tex**

Figure 6: **Qualitative comparisons on ShapeNet car.** Our approach synthesizes more view-consistent and higher-quality textures for fine-grained categories compared to the baseline.

| w/ Resampling | w/ Training | w/Perf.Loss | w/ View Refine | LPIPS ↓ | FID ↓ |
|:---:|:---:|:---:|:---:|:---:|:---:|
| ✓ | ✗ | ✗ | ✗ | 38.89 | 30.924 |
| ✓ | ✓ | ✗ | ✗ | 39.85 | 53.855 |
| ✓ | ✓ | ✓ | ✗ | 38.00 | 30.567 |
| ✓ | ✓ | ✓ | ✓ | **37.89** | **29.998** |

Table 2: Table of Ablation Studies.

view-consistency. Specifically, some methods based on autoregressive processes for texture synthesis exhibit a gradual shift in texture properties (see Fig. 1), leading to reduced view consistency overall. Other approaches that employ score distillation sampling loss (SDS loss) enhance texture quality by optimizing networks that generate texture maps or the texture maps themselves. However, they cannot fully overcome the inherent limitations of SDS loss, such as oversaturation, over-smoothing, and low diversity (Wang et al., 2024; Poole et al., 2022). For example, in the case of "a hammer" (second column in Fig.5), our method successfully textures both the handle and head of the hammer mesh without a gradual shift in each part's texture properties, and avoiding artifacts across multiple viewpoints. These limitations in the other baselines are also reflected in the quantitative results in Tab. 1. In Fig. 6, due to optimization problems with the differential renderer, we only report the texture results for ShapeNet cars using our method and Text2Tex. Notably, even with fine-grained categories, our results show high-quality, view-consistent textures without the aforementioned problems. Additional qualitative comparisons for texture synthesis on Objaverse dataset objects and human meshes can be found in the appendix.

## 4.2 ABLATION STUDIES

**Effects of resampling.** First, we incorporate the resampling method into our baseline, as shown in Fig. 7 (a). The resampling method better preserves the texture properties of the chocolate-covered cherry and muffin liner from the first viewpoint ($v_1$) at the subsequent viewpoints, $v_2$ and $v_4$, compared to the baseline. However, we still observe a gradual change in the pattern of the muffin liner from $v_2$ to $v_4$, and this change becomes more pronounced at $v_{10}$, which is far from $v_1$, in left example of Fig. 7 (b), compromising view-consistency. This tendency is clearly reflected in the results shown in Tab. 2.

**Effects of training with resampled texture.** To address this problem, we fine-tune ControlNet on the resampled textures from five viewpoints ($v_1$, $v_2$, $v_3$, $v_4$, $v_5$) with the performance preservation loss, as shown in Fig. 7 (b). As a result, the ability to preserve texture properties is significantly improved. We now observe that the muffin textures from $v_{11}$—a viewpoint diametrically opposite to $v_1$ with respect to the origin—are similar to the textures from $v_1$.

**Effects of adaptive view refinement.** In Fig. 7(c), we illustrate the texture synthesis process from the side view to the bottom view of the muffin mesh. If the user does not define an appropriate mid-

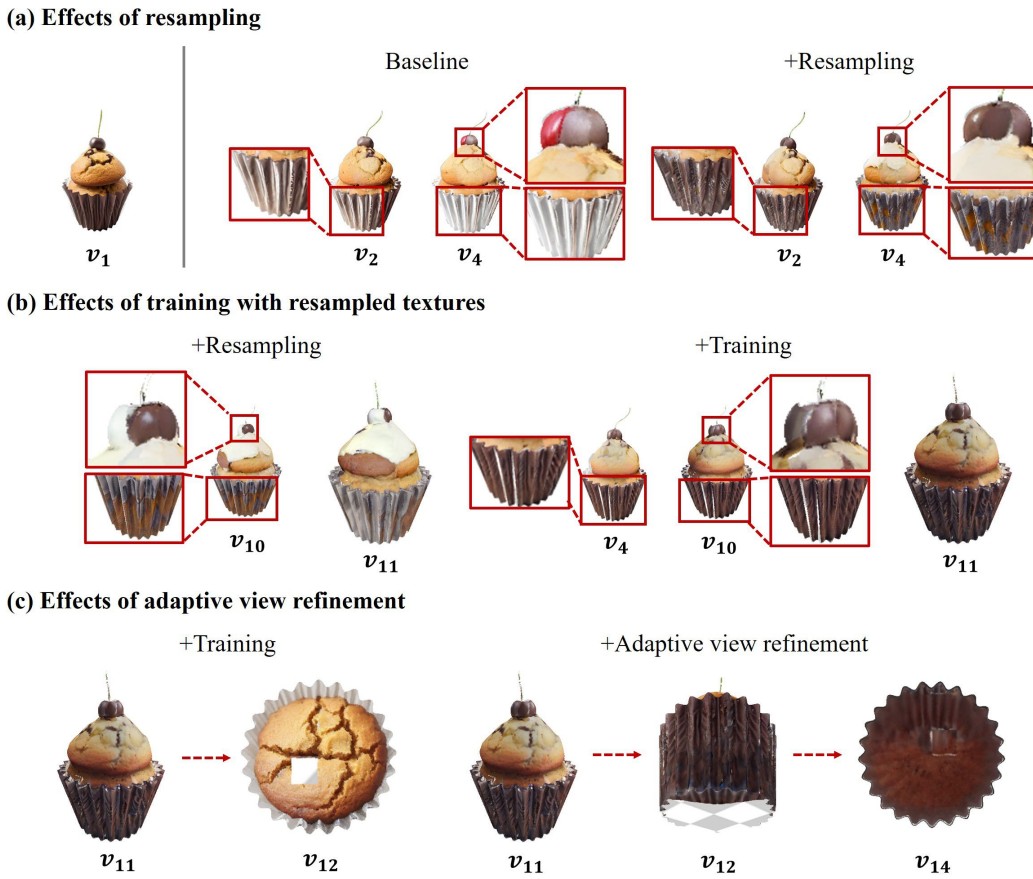

Figure 7: **Ablation studies.** The illustration demonstrates the effectiveness of the key components of TexTailor. As each component is applied, the muffin's texture becomes more consistent across multiple viewpoints, and the overall texture quality improves.

point between these views, as shown in the left example, the model generates a texture that is entirely misaligned with the previously synthesized one in terms of geometry. Conversely, proper compensation at predefined viewpoints effectively prevents this phenomenon. Our method eliminates the need for such cumbersome manual efforts, including tracking and precisely defining camera positions for different geometries.

## 5 CONCLUSION

In this paper, we propose TexTailor, a novel method for synthesizing high-quality textures based on language cues for 3D meshes across multiple categories. Our methodology utilizes a resampling scheme to successfully harmonize unknown regions with known regions in inpainting techniques, thereby improving view consistency between adjacent viewpoints. To address the gradual shift in texture properties, we fine-tune a small set of resampled texture images using ControlNet with an performance preservation loss. Additionally, the adaptive view refinement technique enhances quality and view consistency by leveraging previously synthesized textures, eliminating the need for manual coordination of camera positions based on the object's geometry. Experiments on datasets from various categories demonstrate the superior performance of our approach in synthesizing high-quality, view-consistent textures.

ACKNOWLEDGEMENTS

This research has been supported by the LG Electronics Corporation. (Project No. G01230381)

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

# A APPENDIX

## A.1 ADDITIONAL QUALITATIVE COMPARISON

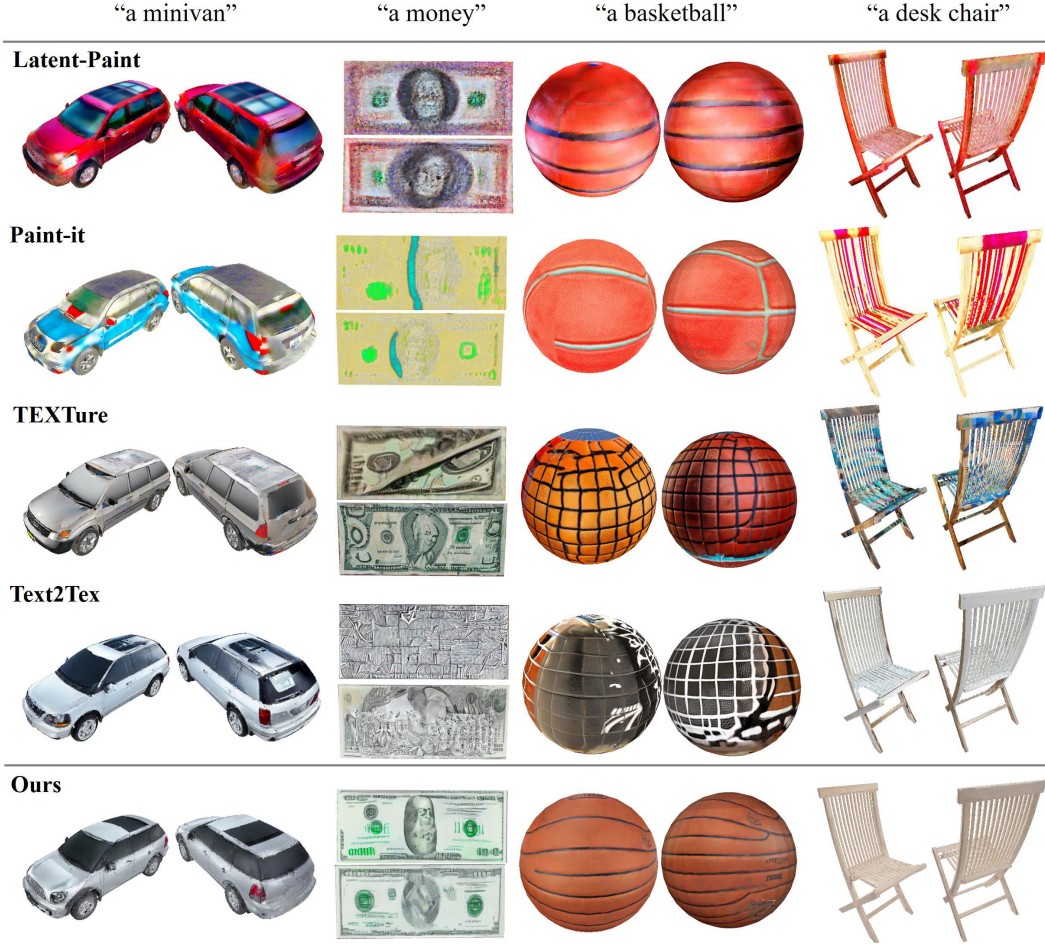

Figure 8: **Additional qualitative comparisons on Objaverse.** We compare TexTailor with state-of-the-art baselines (Metzer et al., 2023; Chen et al., 2023a; Richardson et al., 2023; Youwang et al., 2024) on Objaverse meshes. Compared to other methods, TexTailor produces textures that are more view-consistent and better aligned with object geometries.

**Objaverse Dataset**    To further validate the effectiveness of our proposed methodology, we provide qualitative comparisons for four additional objects from the Objaverse dataset, complementing the four objects already showcased in the qualitative results of the paper (see Fig. 5). Specifically, methods that rely on auto-regressive processes for texture synthesis often encounter a gradual shift in texture properties, leading to the emergence of texture seams that significantly degrade overall texture quality across multiple viewpoints. For instance, in the cases of Text2Tex (Chen et al., 2023a) and TEXTure (Richardson et al., 2023) with the prompt "a basketball", the texture patterns, colors, and materials of the basketball are inconsistently generated, resulting in noticeable variations within a single viewpoint. Similarly, with the prompt "a desk chair," the chair exhibits patches of paint in colors such as white or blue, further emphasizing these inconsistencies.

In contrast, approaches based on SDS loss face challenges stemming from intrinsic limitations of the loss itself, such as oversaturation and low diversity. For example, while Latent-Paint (Metzer et al., 2023) generates textures that appear relatively well-synthesized, it frequently displays a strong bias toward red-dominated textures. Similarly, Paint-it (Youwang et al., 2024) often suffers from visible oversaturation artifacts, detracting from the overall realism of the generated textures.

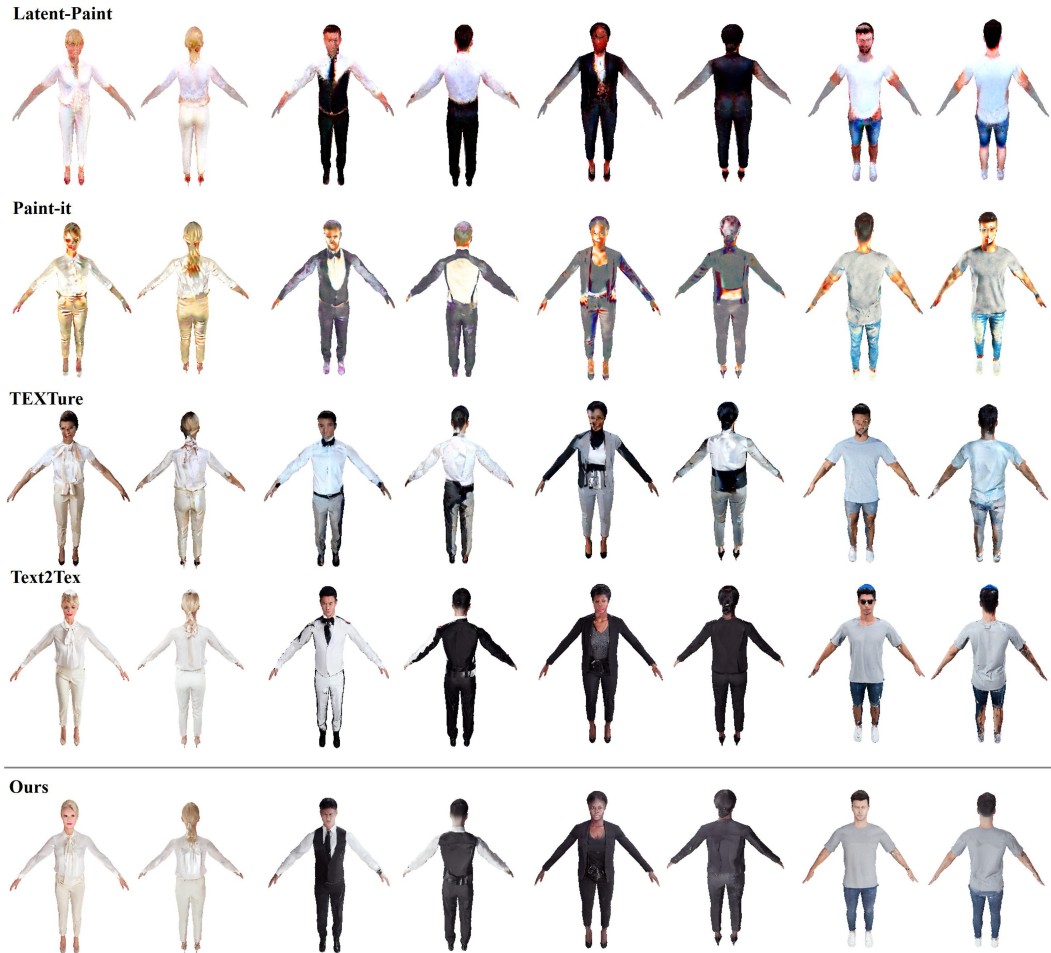

Figure 9: **Additional qualitative comparisons on RenderPeople (ren, 2023).** Qualitative comparison of generated textures for clothed human meshes using various text prompts. Each column corresponds to a different text prompt: (1st col) 'A woman wearing a white blouse with a ribbon detail, light beige pants, nude-tone heels, and neatly tied blonde hair', (2nd col) 'A man, wearing a black vest, black formal trousers, a black tie, a black belt, a white dress shirt, and dark formal shoes, with short neatly styled hair', (3rd col) 'A man, wearing a gray short-sleeve T-shirt, blue jeans, white sneakers, and short, dark brown hair styled neatly', and (4th row) 'A woman with dark skin tone, wearing a black blazer, a black top, gray pants with a gray tied belt, black heels, and having dark hair'. The results demonstrate the performance of our method across diverse prompts and viewpoints.

**Clothed Human Dataset**   we provide qualitative comparison results for clothed human meshes from RenderPeople (ren, 2023). The text prompts for RenderPeople's clothed human meshes were created based on the ground truth images provided by RenderPeople, which were designed by professional designers. Texture synthesis for human meshes demonstrates the superiority of our approach in terms of consistency and quality. For instance, when examining the texture results for the male mesh corresponding to the prompt in the second column - 'A man, wearing a black vest, black formal trousers, a black tie, a black belt, a white dress shirt, and dark formal shoes, with short neatly styled hair' - other methods exhibit a phenomenon of textual gradual shift. This causes the clothing colors shifting inconsistently between white, black, and gray, making it challenging for the models to faithfully reproduce the clothing described in the prompt. In contrast, TexTailor effectively eliminates this issue, accurately representing the prompt with textures like "a black vest" and "a white dress shirt", ensuring precise alignment with the described attire. In addition to clothing, the head region also presents challenges. For instance, in the top view, textures applied to other regions (such as the face, upper body, or lower body) are often not visible, leading to a decline in texture quality for the top view. However, TexTailor addresses this issue by utilizing the adaptive view refinement

technique, which transfers texture information from regions like the face and clothing. This ensures seamless texture generation without visible texture seams, even in challenging viewpoints like the top or bottom view.

## A.2 ADDITIONAL QUALITATIVE RESULTS

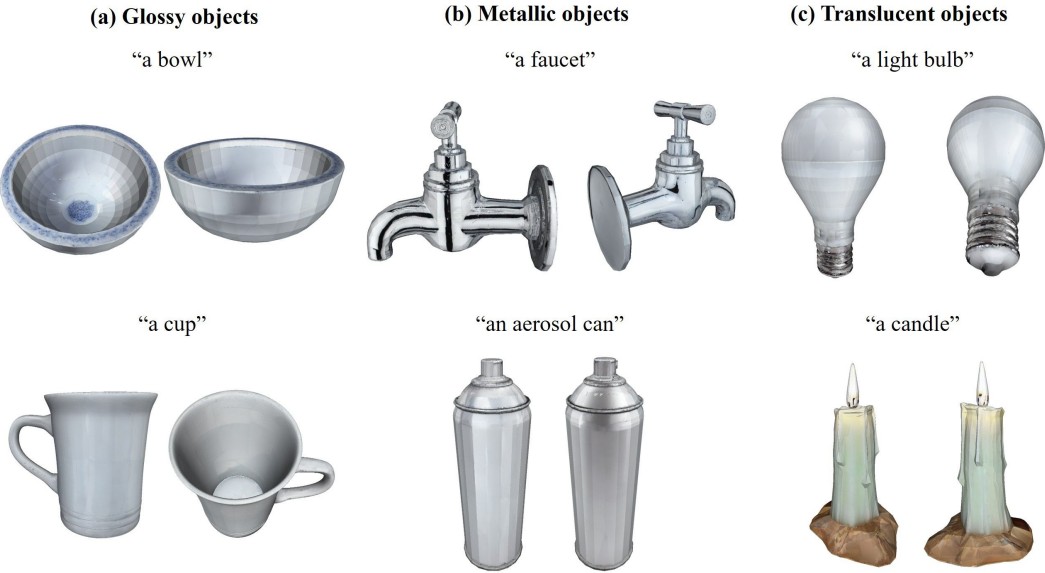

Figure 10: **Additional qualitative comparisons on non-diffuse objects from the Objaverse dataset.** We present the texture synthesis results categorized into three types: (a) Glossy objects, (b) Metallic objects, and (c) Translucent objects.

We further demonstrate the texture quality of not only diffuse objects but also non-diffuse objects from the Objaverse dataset, categorized into three types: (a) glossy objects, (b) metallic objects, and (c) translucent objects. The results generally show satisfactory texture quality across all categories. However, for regions where light reflection or shadows occur, these features are pre-generated based on the synthesis process rather than dynamically determined by the lighting direction in the user's graphics software. This could pose challenges for practical applications. Nevertheless, this limitation is not unique to our approach and is observed in other methods as well. Developing techniques to prevent such biases in light effects during the texture synthesis stage will be a focus of our future work.

## A.3 ADDITIONAL ABLATION STUDIES

**Effects of resampling.** When observing the texture of "a basket," the transition from $v_0$ to $v_1$ reveals a change in the texture of the inside of the basket in the left part. This occurs in the baseline method because the previously synthesized texture visible from the current viewpoint is only merged once per timestep during the diffusion process. In contrast, our methodology improves this by utilizing the resampling strategy proposed in (Lugmayr et al., 2022), which allows the texture to be merged multiple times per timestep, resulting in a more consistent outcome.

**Effects of training with resampled texture.** When observing the object "a briefcase," applying the resampling method generates consistent textures for viewpoints that are close to the initial viewpoint $v_0$, such as the nearby viewpoint $v_5$. However, as the viewpoint moves further away from $v_0$, variations in texture properties begin to appear, and at the opposite viewpoint $v_9$, the texture properties have completely changed. In contrast, when the depth-aware T2I model is fine-tuned using five resampled texture images from viewpoints adjacent to $v_0$ with performance preservation loss, the model fits to the distribution of the resampled textures, leading to noticeable improvements in consistency.

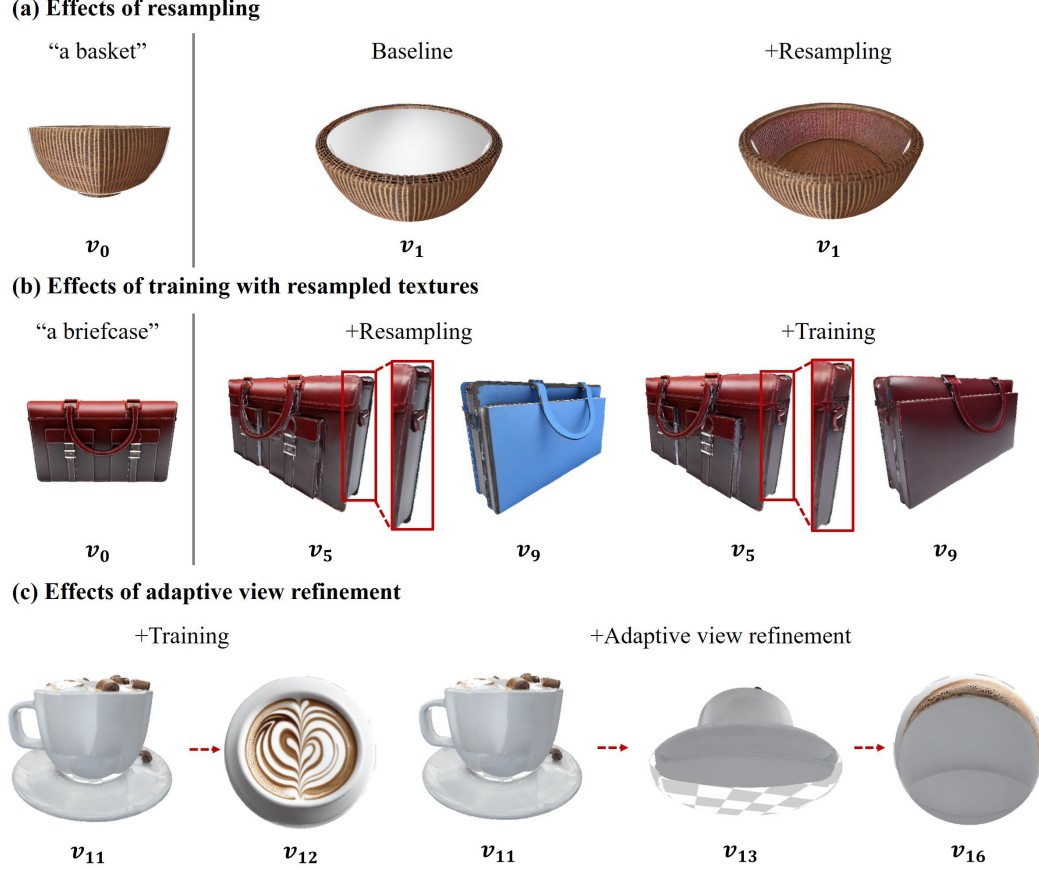

Figure 11: **Ablation studies.** The illustration highlights the effectiveness of the key components of TexTailor. As each component is applied, the textures of various objects become increasingly consistent across multiple viewpoints, demonstrating improved texture quality regardless of the object type.

**Effects of adaptive view refinement.** When observing the object "a cappuccino", transitioning from $v_{11}$ to $v_{12}$ on the left part reveals an issue where the bottom of the cup is generated incorrectly. Since there is no texture information from the previous viewpoint, the model generates the top of the cappuccino instead of the bottom. However, by using the adaptive view refinement technique on the right part, an intermediate viewpoint ($v_{13}$) is automatically added. This not only provides the texture information from $v_{11}$ to guide a more natural texture synthesis but also eliminates the need for the tedious process of manually configuring optimal camera positions.

### A.4 LIMITATIONS

TexTailor faces several limitations. First, the overall texture quality heavily relies on the quality of the five images used to fine-tune the depth-aware T2I diffusion model. Even with the resampling method, certain object angles, even from viewpoints close to the initial one, may still produce textures with inconsistent properties or images misaligned with the depth condition. This reliance on suboptimal training images can sometimes degrade texture quality rather than enhance it. Additionally, patterns frequently observed in viewpoints near the initial one often repeat across the object, leading to unrealistic and repetitive textures. For instance, in the case of an alarm clock, the clock hands learned from the initial viewpoint might appear as texture patterns on the sides or even the back. Furthermore, TexTailor's fine-tuning process is time-intensive, requiring approximately an hour and a half per 3D mesh on an NVIDIA TITAN RTX. Finally, LPIPS does not adequately capture consistency across multiple viewpoints due to spatial misalignments in overlapping sections between adjacent views. To address these limitations, integrating lightweight fine-tuning approaches

such as LoRA (Hu et al., 2021) and developing improved metrics for measuring consistency between arbitrary pairs of views offer promising directions for future work.

## A.5 HIGHLIGHTING DIFFERENCES: A ZOOMED-IN VIEW

**Text2Tex**

**Ours**

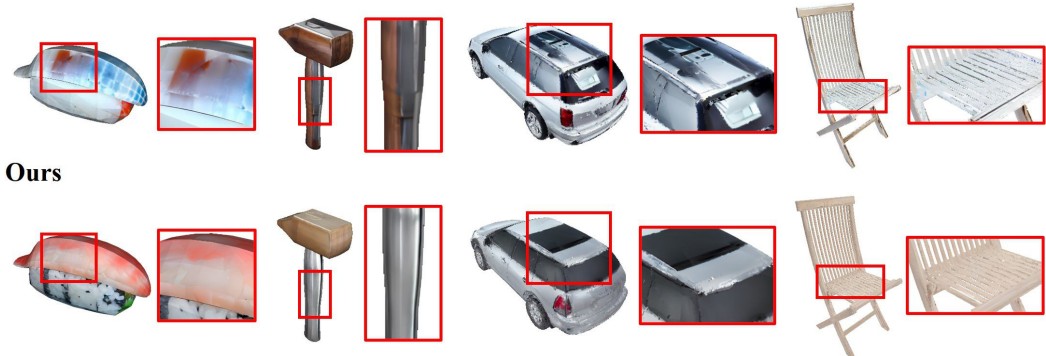

Figure 12: **Zoomed-in qualitative comparisons between TexTailor and the baseline Text2Tex** The red boxes highlight regions where visual differences in texture consistency and quality are more apparent, helping to illustrate the effectiveness of TexTailor in preserving texture properties and minimizing artifacts.

We aim to enhance the clarity of the qualitative comparisons between TexTailor and the baseline Text2Tex by providing zoomed-in patches for four challenging cases ("a sushi," "a hammer," "a minivan," and "a desk chair") where visual differences are harder to discern in the qualitatve comparison sections(Sec. 5 and Sec. 8).

## A.6 ABLATION STUDY ON THE THRESHOLD $\beta$

We compare our results in Tab. 3 by varying the threshold $\beta$ value from Eq. 12. For this experiment, we randomly select 100 meshes from the Objaverse dataset, ensuring no overlapping categories among the 400 meshes, and evaluate them using FID and LPIPS metrics. When the threshold is set too low (e.g., 0.25), the information for the keep region is not properly reflected, resulting in lower scores. Conversely, when the threshold is set too high (e.g., 0.75), objects tend to be painted in overly fragmented parts rather than as cohesive units. This intensifies the gradual shift effect, also leading to lower scores. Based on these observations, we set $\beta$ to 0.5, a balanced value, for all experiments in this paper.

Table 3: Quantitative effect of varying threshold $\beta$ on the adaptive view refinement technique.

| $\beta$ | 0.25 | 0.5 | 0.75 |
|---|---|---|---|
| LPIPS ↓ | 8.99464 | **8.99176** | 8.99468 |
| FID ↓ | 58.567 | **57.799** | 59.503 |

