# OpenReview forum: "TexTailor: Customized Text-aligned Texturing via Effective Resampling"
_ICLR.cc/2025/Conference — ICLR 2025 Poster_

### Official Review · Reviewer_NBoV · 2024-10-21

**Soundness:** 3
**Presentation:** 3
**Contribution:** 2
**Rating:** 6
**Confidence:** 5

**Summary:**

This paper proposed a novel architecture that can generate more consistent 3D texture than TEXTure and Text2Tex.

**Strengths:**

1. Proposed a better approach for viewpoint sampling.
2. The performance of the proposed method is better than listed SOTAs.

**Weaknesses:**

1. The resampling strategy is similar to the [TexFusion:Synthesizing 3D Textures with Text-Guided Image Diffusion Models] and [TexGen: Text-Guided 3D Texture Generation with Multi-view Sampling and Resampling]. Please explain the difference.
2. In Fig. 1b, it seems that the proposed method is over-smoothed. Please explain the reason.
3. Please answer the following questions.

**Questions:**

1. I do not get the correlation of the third paragraph in Section I with the main content. I think the geometry conversion is not a problem to be solved in this paper. And the SDS can be directly applied on the mesh (DMTET) which does not need conversion.
2. The authors mentioned that " we can achieve high-quality texture with only 30 steps, significantly fewer than the 250 steps required by the original resampling method for a single view." Other methods like texture and text2tex only sampled no more than 50 steps for each view. Where is this "250" from.
3. What is the meaning of resampling steps? Does it mean you have to sample R steps for each view at each timestep?
4. The authors used "resampled images near the first viewpoint to extract images of the same object from different angles in the output domain of the diffusion model" . How to make sure that the viewpoints near the first viewpoint maintain the same style as the first view.
5. In the loss function of Eqn. 10, the target is constraining the new noise estimation to be the same as the original noise estimation. The what is the meaning of training? The optimal case is keeping the original model unchanged.
6. Any 3D results of the method? I prefer to see the rendered 360-degree videos of results.
7. The attention feature injection as in [Text-Guided Texturing by Synchronized Multi-View Diffusion] can help to reduce the problem of the autoregressive inpainting. Have you tried this?

---

> ### Author Response · Authors · 2024-11-23
>
> Thank you for taking the time to review our work. We greatly appreciate your recognition of the key aspects of our proposed methods and your positive feedback on the experiments. We will revise the manuscript to incorporate your suggestions. If any part of our response is unclear, please do not hesitate to reach out for further clarification. Before addressing the main points of the rebuttal, we have standardized the notation to align with the terminology and references used in our submitted manuscript, rather than those from the cited papers. We kindly ask for your understanding regarding this adjustment.
>
> Detailed response to comments:
> > Q1. The resampling strategy is similar to the [TexFusion:Synthesizing 3D Textures with Text-Guided Image Diffusion Models] [1] and [TexGen: Text-Guided 3D Texture Generation with Multi-view Sampling and Resampling] [2]. Please explain the difference.
>
> In TexFusion [1], the method progressively transitions between viewpoints during the diffusion process, denoising the areas corresponding to "new" regions at each timestep. When transitioning from one viewpoint to another, inconsistencies in noise levels may arise between the "keep" and "new" regions. To resolve this, TexFusion [1] adds an additional step of noise to the "keep" region to align the noise levels. Over time, after completing this process across multiple viewpoints at each timestep, the areas corresponding to "new" regions are aggregated into a single texture map for that timestep. Consequently, the entire object is textured across all viewpoints, with denoising occurring once per timestep from simple Gaussian noise.
>
> In contrast, our method differs in that, within each timestep of the diffusion process, the process of adding noise and removing it (denoising) is performed multiple times for the "new" region while simultaneously merging it with the "keep" region. This process is repeated $R$ times per timestep.
>
> On the other hand, the "resampling" described in TexGen [2] differs from our definition of resampling. In TexGen [2], resampling involves estimating $z_0$, the final denoised result at each timestep, and using this estimate to predict the noise network's value at the same step. In other words, their resampling strategy focuses on re-sampling the noise network value ${\epsilon}_{\phi}$. In contrast, our method's resampling strategy involves re-sampling the latent vector $z_t$ multiple times.
>
> In summary, compared to TexFusion [1], our method differs in that we perform the process of adding noise and removing it (denoising) multiple times within each timestep (i.e., a difference in the number of sampling iterations at each time step). Furthermore, our resampling approach differs from that of TexGen[2] in both the target (resampling ${\epsilon}_{\phi}$ versus $z_t$) and the method employed.
>
> > Q2. In Fig. 1b, it seems that the proposed method is over-smoothed. Please explain the reason.
>
> If the depth-aware diffusion model lacks sufficient training for a specific viewpoint of the object image conditioned on text descriptions or depth maps, or if the texture information applied from previous viewpoints (corresponding to the "keep" regions in Sec. 2.2) is insufficient to generate consistent textures in textureless areas (corresponding to the "new" regions in Sec. 2.2) due to geometric constraints (e.g. objects that appear small from a specific angle), inference for that particular angle while aligning texture properties from the initial viewpoint becomes challenging. While our methodology ensures that the object's properties are preserved, this limitation may lead to a slight loss of texture detail, resulting in an over-smoothed appearance. However, as demonstrated in Text2tex [3] and Texture[4], this issue is effectively mitigated through an update process (Sec.2.2) that refines the detailed regions.

---

> ### Author Response · Authors · 2024-11-23
>
> > Q3. I do not get the correlation of the third paragraph in Section I with the main content. I think the geometry conversion is not a problem to be solved in this paper. And the SDS can be directly applied on the mesh (DMTET) which does not need conversion.
>
> This paper focuses on texture map generation based on explicit representations in 3D modeling. To emphasize this, the third paragraph discusses the advantages of explicit representations over implicit ones, such as their ease of integration into graphics engines and real-time applications. However, as noted in the review, the explanation regarding the necessity of generating texture maps within explicit representations may have been somewhat unclear. With your permission, I propose revising this section to provide greater clarity as follows:
>
> "While these methodologies provide both geometry and texture, converting implicit neural representations into explicit formats, such as meshes, remains necessary for integration into graphics engines and real-time applications. Recently, DMTet [5] has enabled precise mesh geometry extraction from implicit representations by leveraging a signed distance field and the Marching Tetrahedra algorithm. However, in texture synthesis, texture unwrapping often leads to inconsistent mappings, which can degrade the visual quality of the output or necessitate additional texture synthesis steps.[6][7]"
>
> > Q4. The authors mentioned that " we can achieve high-quality texture with only 30 steps, significantly fewer than the 250 steps required by the original resampling method for a single view." Other methods like texture and text2tex only sampled no more than 50 steps for each view. Where is this "250" from.
>
> Lugmayr et al. (2022) [8], who first proposed the resampling method for denoising diffusion probabilistic models, stated that 250 timesteps are required to achieve a proper inpainting effect with their resampling strategy. In contrast, by incorporating the DDIM scheme in our methodology, we reduced the number of timesteps to 30 without compromising quality.
>
> > Q5. What is the meaning of resampling steps? Does it mean you have to sample R steps for each view at each timestep?
>
> That's correct. What I meant is that noise addition and removal are repeated $R$ times at each timestep. I apologize for the confusion. Do you think changing the name from "resampling" to "multiple resampling" might make it easier to understand?
>
> > Q6. The authors used "resampled images near the first viewpoint to extract images of the same object from different angles in the output domain of the diffusion model". How to make sure that the viewpoints near the first viewpoint maintain the same style as the first view.
>
> For viewpoints close to the first viewpoint, the texture applied from the first viewpoint is repeatedly conditioned into the diffusion process through the resampling strategy. This increases the likelihood of generating textures in a style similar to the first viewpoint. Furthermore, for the five viewpoints nearest to the first viewpoint, we assumed that the textures would be generated in the same style as the first viewpoint. This assumption is based on the fact that, at the end of processing each of these five viewpoints, ControlNet is trained using the resampled texture images synthesized from the previous viewpoints, ensuring consistency across the textures generated for these five viewpoints.
>
> > Q7. In the loss function of Eqn. 10, the target is constraining the new noise estimation to be the same as the original noise estimation. The what is the meaning of training? The optimal case is keeping the original model unchanged.
>
> Our final loss is defined in Eq. (11) by combining Eq. (9) and Eq. (10). Here, $\epsilon$ and $\epsilon_{\text{orig}}$ are different, so the optimal case cannot simply be assumed to preserve the parameters of the original model. The reason for introducing Eq. (10) is that, when training ControlNet with a limited number of images (in this case, rendered images from viewpoints close to the first viewpoint), there is a high risk of the model forgetting the information it had previously learned. This leads to a phenomenon known as catastrophic forgetting, which can impair ControlNet's capabilities. To prevent this, Eq. (10) ensures that the parameters of ControlNet remain close to the values learned during its original training with a large dataset.

---

> ### Author Response · Authors · 2024-11-23
>
> > Q8. The attention feature injection as in [Text-Guided Texturing by Synchronized Multi-View Diffusion] can help to reduce the problem of the autoregressive inpainting. Have you tried this?
>
> If you’re referring to the Self-Attention Reuse part proposed in Text-Guided Texturing by Synchronized Multi-View Diffusion [9], unfortunately, I hadn’t considered this methodology before submitting the paper and therefore have not attempted it. However, after reading the paper, I believe that utilizing the self-attention blocks of ControlNet’s encoder during the training process could be a novel and promising approach. Currently, due to a lack of time and resources, we are unable to reproduce all the results, but I will definitely consider trying it as part of my future work. Thank you for recommending such an excellent paper.
>
> > Q9. Any 3D results of the method? I prefer to see the rendered 360-degree videos of results.
>
> In addition to the four objects from the Objaverse dataset presented in the qualitative results of the paper, we will showcase qualitative comparisons for four additional objects from the same dataset in GIF format. Additionally, we will provide qualitative comparison results for clothed human meshes from RenderPeople[10], also presented in GIF format.
>
> The text prompts for RenderPeople's clothed human meshes were created based on the ground truth images provided by RenderPeople, which were designed by professional designers.

---

> ### Author Response · Authors · 2024-11-23
>
> #### 1.Objaverse Dataset
> |Text Prompt|Latent-Paint|Text2Tex|TEXTure|Paint-it|Ours|
> |:-----:|:-----:|:-----:|:-----:|:-----:|:-----:|
> |"a basketball"|[link](https://drive.google.com/file/d/1PUhh1CFVGfzvHqeNS7-VWlspWygV7M9i/view?usp=drive_link)|[link](https://drive.google.com/file/d/10-Vjeq2lBFIrApNeJ409yoJL3tbdL7M5/view?usp=drive_link)|[link](https://drive.google.com/file/d/1-zmxggvpEi17NlXk4rwqswcguWHToUnx/view?usp=drive_link)|[link](https://drive.google.com/file/d/1tGnNDX-1Z1AWRACQzloKAnW7EJrbkIgV/view?usp=drive_link)|[link](https://drive.google.com/file/d/1ChOOEJCHugwlXSTZMtRPVyR7MOBsXiow/view?usp=drive_link)|
> |"a cd player"|[link](https://drive.google.com/file/d/1fKhj-8aeGkVKjfb3FwtGUBkuf7FuKiM2/view?usp=sharing)|[link](https://drive.google.com/file/d/1VdXOQslsVru-x2x19tGYfKttT9rI7hit/view?usp=sharing)|[link](https://drive.google.com/file/d/1JE_LDcZm1-oBX4QJN-dIlmBCcNzJz2ne/view?usp=sharing)|[link](https://drive.google.com/file/d/1sRkv6iXBa7e9TVSMcV3hrlbZ8tZo0u_m/view?usp=sharing)|[link](https://drive.google.com/file/d/17k4XOBHya95ZsYwqRs_NONUff9Apw7r4/view?usp=sharing)|
> |"a desk chair"|[link](https://drive.google.com/file/d/1mm_0cvW2RsvktsLcy07JXAeS5CGVfENO/view?usp=drive_link)|[link](https://drive.google.com/file/d/1yJQkmI6THlNn3hoL-mW2O7D5CI_A6gq9/view?usp=drive_link)|[link](https://drive.google.com/file/d/1pRvTW1eEjVNk7rycoJRAJ8IdvPqHEj46/view?usp=drive_link)|[link](https://drive.google.com/file/d/1Jpx9vtBIV7ymD7zRHLBIPSNAxSItMjFe/view?usp=drive_link)|[link](https://drive.google.com/file/d/1ELVqtUhQbT5XmUcqko0h1pJjMGaicmAl/view?usp=drive_link)|
> |"a dumpster"|[link](https://drive.google.com/file/d/1HKaoGX_dUN0-TQXlmIAuCbHygUFUV3bD/view?usp=drive_link)|[link](https://drive.google.com/file/d/1f1m09_mCKa9_4KZOGKme8BwsAc_7lkJG/view?usp=drive_link)|[link](https://drive.google.com/file/d/18UZWI-sGhUqXNuFW0SdnIFFO_QNVoiNI/view?usp=drive_link)|[link](https://drive.google.com/file/d/1zN_VombixmEsueJfd6L5kP7zws5fzq0G/view?usp=drive_link)|[link](https://drive.google.com/file/d/1jtiXfphExSoW4wGiDGUBRTD1t4nGVdKi/view?usp=drive_link)|
> |"a hammer"|[link](https://drive.google.com/file/d/1RSq-vzcPfclttJToHObHypn7hs17VLEd/view?usp=sharing)|[link](https://drive.google.com/file/d/1Kw7aEhSdx3mCXdYkGKRzrGeobqaLJwAW/view?usp=sharing)|[link](https://drive.google.com/file/d/1aexhUd7Nl0Aey3Vg4I-XnoAdlzb3UZPJ/view?usp=drive_link)|[link](https://drive.google.com/file/d/1kxIuyKW3ROjT25KaohiRxpwVAwvQaVWC/view?usp=sharing)|[link](https://drive.google.com/file/d/1GqPePJEubcHmS5LOnMir5OmVIsHv8iTL/view?usp=drive_link)|
> |"a minivan"|[link](https://drive.google.com/file/d/1yEG9Ni4rrNaZ4kCvL4WRgrY5nU2CxfZS/view?usp=drive_link)|[link](https://drive.google.com/file/d/11nAvcQqOKF-fCaP5-ZAHBNTzD27Fujz2/view?usp=drive_link)|[link](https://drive.google.com/file/d/1sUQbNuGWEw9S85RozmnzII0ByIneFv3W/view?usp=drive_link)|[link](https://drive.google.com/file/d/1ku15fjR4ts_2KJm0Rl6OqGi5h9mP3hqi/view?usp=drive_link)|[link](https://drive.google.com/file/d/1EJcNOro3_wlw6spZ7TYFwtWfZ1Vx3dbL/view?usp=drive_link)|
> |"a money"|[front](https://drive.google.com/file/d/10rPPgZsWc0iotldevklujifOZMbiz-0X/view?usp=drive_link), [back](https://drive.google.com/file/d/1PaRrWMzpS073RU7bg8NKvnB2wVvnF1ZX/view?usp=drive_link)|[front](https://drive.google.com/file/d/1uNwXkh7O4UASFmm_BhDJJfX8MYWj_My6/view?usp=drive_link), [back](https://drive.google.com/file/d/10x-gy15J7OgJdymqFsalM_R-aF-XCJi4/view?usp=drive_link)|[front](https://drive.google.com/file/d/1rp4m1AdduMMOUyXMHUng5eq8jmQBsF1e/view?usp=drive_link), [back](https://drive.google.com/file/d/1mRWQlFv7YF4SeOrPSq4GrXkro6nveYzC/view?usp=drive_link)|[front](https://drive.google.com/file/d/1EFDwk-csQEiokhU09NiubX-yGC4NNXr7/view?usp=drive_link), [back](https://drive.google.com/file/d/1I_uEbvFRMn9HdfhjgDSNf7miW_fgTLQf/view?usp=drive_link)|[front](https://drive.google.com/file/d/1SRVIlpRovqU2o2d1XYQMkdHXsgd_TQxC/view?usp=drive_link), [back](https://drive.google.com/file/d/1Oc45Uj5M5ySbKikOYqvl6jA0KX9sEB6O/view?usp=drive_link)|
> |"a sushi"|[link](https://drive.google.com/file/d/15QWzVwVvJ-r7rFqUcVXGhnmNGRMdUClx/view?usp=drive_link)|[link](https://drive.google.com/file/d/1d7hxtTi8NFio4kAXeMJjIGyrIaOwmfmA/view?usp=drive_link)|[link](https://drive.google.com/file/d/1uZerJAnPSwV-bAw0e3JXwBpaPsL96XzK/view?usp=drive_link)|[link](https://drive.google.com/file/d/1VMeuBZoB-CuDDc0ekSCb5ezrXQLeVNgr/view?usp=drive_link)|[link](https://drive.google.com/file/d/1gSCl1HYbhxIK4ovEEs0OJVF1CgqIgHLg/view?usp=drive_link)|

---

> ### Author Response · Authors · 2024-11-23
>
> #### 2.RenderPeople Dataset
> |Text Prompt|Latent-Paint|Text2Tex|TEXTure|Paint-it|Ours|
> |:-----:|:-----:|:-----:|:-----:|:-----:|:-----:|
> |"A business woman wearing a white blouse with a ribbon detail, light beige pants, nude-tone heels, and neatly tied blonde hair"|[claudia](https://drive.google.com/file/d/1cugJNiHxkIxzZmfapOkUSUt1b5nmX-I4/view?usp=drive_link)|[claudia](https://drive.google.com/file/d/1tuUSCLRtZs8meKal50_kkfX0XkTDIlwU/view?usp=drive_link)|[claudia](https://drive.google.com/file/d/1HkO7HNKWkFh4u_xZfxRTgiyQV35ENyvS/view?usp=drive_link)|[claudia](https://drive.google.com/file/d/1cYYHgVoRn5YN-BQGbSXxtVa3ONWLBpIU/view?usp=drive_link)|[claudia](https://drive.google.com/file/d/1jpUILlxb39eSsAhZnk0kMBt2dTQ_iswl/view?usp=drive_link)|
> |"A man, wearing a white dress shirt, a black vest, black formal trousers, a black tie, a black belt, and dark formal shoes, with short neatly styled hair"|[eric](https://drive.google.com/file/d/1zu8JFHMlV2oHCu84je9z-hJn1_YUACl0/view?usp=drive_link)|[eric](https://drive.google.com/file/d/1ejcq1bB6djVUmmCqAxbqrDA2q_l_mOsK/view?usp=drive_link)|[eric](https://drive.google.com/file/d/1kQ4JW-Nu14p8qiLvR4RwvB90Iu_elRoI/view?usp=drive_link)|[eric](https://drive.google.com/file/d/1FItS8NhnffnzjoNPPU1or_t6ScWUveO2/view?usp=drive_link)|[eric](https://drive.google.com/file/d/1YibXGY7AicDrCaalaJUChKGRE5nQ6T7c/view?usp=drive_link)|
> |"A man, wearing a gray short-sleeve T-shirt, blue jeans, white sneakers, and short, dark brown hair styled neatly"|[manuel](https://drive.google.com/file/d/1yRmwmkkZ8c2VmC3fr2B92R2ztWT6BC0a/view?usp=sharing)|[manuel](https://drive.google.com/file/d/1jasYCugvfEaWtXMusmCFExluKTggHtKX/view?usp=sharing)|[manuel](https://drive.google.com/file/d/1kXzUleXzPhJkehtuoNUPhcj9YJzqYSRm/view?usp=sharing)|[manuel](https://drive.google.com/file/d/1ZYkWlhq2cxa-jyxaPeHbdvqTmjQE9-RP/view?usp=sharing)|[manuel](https://drive.google.com/file/d/1epeP2Ms2HC5AkW7uumw9f08H6dk-FVCc/view?usp=sharing)|
> |"A woman with medium-dark skin tone, wearing a black blazer, a black top, gray pants with a gray tied belt, black heels, and having neatly styled dark hair"|[carla](https://drive.google.com/file/d/1SN1Hft8lAbLU4HHTxB7k6e_WxoG-h-r-/view?usp=drive_link)|[carla](https://drive.google.com/file/d/1jJCNNpgXsB8xJKu6NnGkXicQ3ZDUOCRf/view?usp=drive_link)|[carla](https://drive.google.com/file/d/1OZ-uWrrX06M19CDgpukyCMkM0A5A4l78/view?usp=drive_link)|[carla](https://drive.google.com/file/d/1vc3EaAvNO18t8bUo8l_-KC5hhGTOzDTD/view?usp=sharing)|[carla](https://drive.google.com/file/d/1TlZ2wcokdUOz9Q-KIsSIShCiheelB1ox/view?usp=sharing)|
>
>
> ### Reference
> [1] Cao, Tianshi, et al. "Texfusion: Synthesizing 3d textures with text-guided image diffusion models." Proceedings of the IEEE/CVF International Conference on Computer Vision. 2023.
>
> [2] Huo, Dong, et al. "TexGen: Text-Guided 3D Texture Generation with Multi-view Sampling and Resampling." European Conference on Computer Vision. Springer, Cham, 2025.
>
> [3] Chen, Dave Zhenyu, et al. "Text2tex: Text-driven texture synthesis via diffusion models." Proceedings of the IEEE/CVF International Conference on Computer Vision. 2023.
>
> [4] Richardson, Elad, et al. "Texture: Text-guided texturing of 3d shapes." ACM SIGGRAPH 2023 conference proceedings. 2023.
>
> [5] Shen, Tianchang, et al. "Deep marching tetrahedra: a hybrid representation for high-resolution 3d shape synthesis." Advances in Neural Information Processing Systems 34 (2021): 6087-6101.
>
> [6] Lin, Chen-Hsuan, et al. "Magic3d: High-resolution text-to-3d content creation." Proceedings of the IEEE/CVF Conference on Computer Vision and Pattern Recognition. 2023.
>
> [7] Chen, Rui, et al. "Fantasia3d: Disentangling geometry and appearance for high-quality text-to-3d content creation." Proceedings of the IEEE/CVF international conference on computer vision. 2023.
>
> [8] Lugmayr, Andreas, et al. "Repaint: Inpainting using denoising diffusion probabilistic models." Proceedings of the IEEE/CVF conference on computer vision and pattern recognition. 2022.
>
> [9] Liu, Yuxin, et al. "Text-guided texturing by synchronized multi-view diffusion." arXiv preprint arXiv:2311.12891 (2023).
>
> [10] RenderPeople, https://renderpeople.com/free-3d-people/, 2023

---

### Official Review · Reviewer_ujQC · 2024-10-27

**Soundness:** 3
**Presentation:** 1
**Contribution:** 2
**Rating:** 6
**Confidence:** 4

**Summary:**

The paper focuses on **consistent texture synthesis**. The authors analyze the artifacts in the current approaches and propose a new approach, **TexTailor**, to keep synthesized textures consistent across different viewpoints. TexTailor equips with a resampling scheme for integrating previous textures, a finetuned depth-aware T2I model trained with performance preservation loss, and an adaptive viewpoint refinement strategy for inpainting.

The authors evaluate the performance of TexTailor on a subset of Objaverse dataset, and showcases that TexTailor outperforms state-of-the-art methods.

**Strengths:**

## Motivation
The paper starts with an analysis of the limitations of previous methods. It hypothesizes those inconsistent results from previous methods are mainly coming from an inappropriate way of integrating information from previously synthesized textures. Given this agile insight, it tries to addresses the inconsistency issue by proposing a new approach to better use information across different viewpoints and previously synthesized textures.

The motivation of the paper is more about a technical aspect. The analysis of previous approaches makes sense.

## Method
- In Section 3.2, the problem of ControlNet for incorporating multi-views is interesting.
- In Section 3.3, the analysis of setting viewpoints sounds interesting (Line 303-310). Using a proportion (Eqn. (12)) is an intuitive way.

## Experiments
- TexTailor outperforms the previous methods in terms of view consistency and quality, as shown in Table 1.
- The ablation study shows a progressive improvement of each component.

The authors also show the limitation of TexTailor - the processing time could be further improved.

**Weaknesses:**

What concerns me the most in this paper is the motivation behind some technical parts and its unclear writing.

## Motivation
- In Line 93, it is not clear to me why finetuning a depth-aware T2I model matters. Maybe including a brief explanation could be helpful.

## Method
- In Section 3.1, the authors propose a non-Markov process to reduce the sampling steps. However, the benefits of it is confusing to me. Would it involve a faster sampling speed? If it would, there is not result to support it. On the other hand, the authors mainly show the effects of resampling is to "preserve the texture properties" (Line 480). This makes me confused about the motivation of newly proposed resampling trick.

## Experiments
- It does not make sense to me the authors choose to not compare with text-driven methods (Line 373-374) just because they have "difficulties" when optimizing textures for "cars". Wouldn't it be a good chance to showcase the superiority of TexTailor?
- The authors do not show any viewpoint-varying results in video format, making it less convincing that TexTailor achieves a good view consistency.
- It is hard to see obvious improvement from TexTailor in Figure 5, especially comparing with Text2Tex. Perhaps including some bounding boxes and zoom-in patches would help.

## Writing/Delivery
The writing of the papers can be further improved. For example,
- Most of the figures in the paper are compressed, resulting in blurriness and sometimes hard to read.
- In Fig.1, citing previous methods (i.e., Text2tex and Texture) might make readers easier to check the idea of them.
- It is challenging for readers to digest Eqn. (6) - (8). A good strategy to improve it might be similar to what Repaint shows in their paper: demonstrating all the terms in a figure with pictures for a vivid demonstration. Current delivery of newly proposed resampling way in Section 3.1 is hard for readers to understand, especially about the main difference between it and Repaint.
- Fig.3 does not deliver a clear message for each component. For example, simply giving readers two equations does not help them to understand what is going on. It might be helpful if the authors can name these two equations in high level.

**Questions:**

1. What are the difficulties for the text-driven methods mentioned in Lines 373 and 374?
2. Is LPIPS (Section 4.1, Evaluation metrics) a good metric to evaluate view consistency, as LPIPS is sensitive to spatial information? Given that the view angles are known, would it make more sense to reproject one of the views to another and then compute LPIPS between the projected view and the other one?
3. What does the performance preservation loss do in Eqn. (10)? Why would it be effective at a high level?

Some of the questions may have been entangled with the Weaknesses.

---

> ### Author Response · Authors · 2024-11-23
>
> Thank you for taking the time to review our work. We greatly appreciate your recognition of the key aspects of our proposed methods and your positive feedback on the experiments. We will revise the manuscript to incorporate your suggestions. If any part of our response is unclear, please do not hesitate to reach out for further clarification
>
> Detailed response to comments:
> > Q1. In Line 93, it is not clear to me why finetuning a depth-aware T2I model matters. Maybe including a brief explanation could be helpful.
>
> As shown in Fig. 1(b), existing methods sequentially transition from the initial viewpoint to subsequent viewpoints, generating and merging textures using a depth-aware T2I model. However, this approach has two major issues: (1) the textures synthesized at the initial viewpoint become invisible as the viewpoint transitions, and (2) even when using an image inpainting strategy, the generated textures can vary significantly depending on the viewing angle. For example, in the pencil case example of Fig. 1(b) under Text2Tex, the blue texture generated from the second viewpoint may appear natural from that specific angle. However, when observed from the angle shown in the third row, only the blue texture region (with the textures synthesized at the first viewpoint no longer visible) is conditioned into the depth-aware T2I model, leading to the generation of textures that deviate from the original texture properties.
>
> To address these issues, this paper assumes that the inability of depth-aware T2I models to maintain texture properties from the initial viewpoint when transitioning across viewpoints is the root cause of the problem. To mitigate this, we additionally train the depth-aware T2I model using a small set of texture images synthesized from viewpoints close to the first viewpoint (which effectively preserve the texture properties of the first viewpoint). This ensures that the depth-aware T2I model is better fitted to those texture properties, enabling the generation of textures with similar properties even as the viewpoint transitions.
>
> >Q2. In Section 3.1, the authors propose a non-Markov process to reduce the sampling steps. However, the benefits of it is confusing to me. Would it involve a faster sampling speed? If it would, there is not result to support it. On the other hand, the authors mainly show the effects of resampling is to "preserve the texture properties" (Line 480). This makes me confused about the motivation of newly proposed resampling trick.
>
> As you mentioned, the resampling trick plays a key role in improving texture consistency between two adjacent viewpoints. This resampling trick is based on the Lumayr et el.[8] from the 2D image inpainting field, where the authors recommend 250 timesteps for applying the resampling trick.
>
> However, such a high number of timesteps makes it challenging to apply the resampling trick to the texture synthesis domain. Specifically, when performing $R$-step resampling per timestep, the total number of timesteps required in 2D image synthesis is $250 \times R$. In contrast, for 3D texture synthesis, the process must account for the number of viewpoints, leading to $250 \times R \times $(Number of viewpoints) timesteps in total. This results in an exceedingly high computational cost for generating the entire texture.
>
> To address this issue, we replaced the DDPM-based sampling[2] method used in Repaint with a DDIM-based sampling [3] method, which allowed us to reduce the number of required timesteps per viewpoint from 250 to 30 while maintaining high quality. We believe the quality preservation is demonstrated in our ablation study, which shows consistent results with and without the resampling trick.
>
> >Q3. It does not make sense to me the authors choose to not compare with text-driven methods (Line 373-374) just because they have "difficulties" when optimizing textures for "cars". Wouldn't it be a good chance to showcase the superiority of TexTailor? What are the difficulties for the text-driven methods mentioned in Lines 373 and 374? (Weakness3 & Question1)
>
> The "difficulty" mentioned in lines 373–374 of this paper refers to cases where the image generated by the depth-aware T2I model is not properly back-projected onto the corresponding part of the mesh.
>
> |Model|Generated Output|Projecting Output|
> |:----:|:----:|:----:|
> |TEXTure|[link](https://drive.google.com/file/d/1INKUqQvy-zIiO9lraq0Tm-XEkPWogfej/view?usp=drive_link)|[link](https://drive.google.com/file/d/1osU9Ss9M60D2-TNjW1fRIaTVyq8bKhPk/view?usp=drive_link)|
>
> While, as you pointed out, this could be argued as an advantage of xatlas-based texture mapping methods used in Text2Tex and TexTailor (our method), the primary focus of this paper is not to conduct a detailed comparison between xatlas[4] and differential mesh rendering (NVDiffRast[5]) or propose a method based on such an analysis. Therefore, to maintain consistency in the manuscript, we opted to briefly mention this issue and move on.

---

> ### Author Response · Authors · 2024-11-23
>
> >Q4. Is LPIPS (Section 4.1, Evaluation metrics) a good metric to evaluate view consistency, as LPIPS is sensitive to spatial information? Given that the view angles are known, would it make more sense to reproject one of the views to another and then compute LPIPS between the projected view and the other one?
>
> I feel that I might not fully understand this part, so I would like to kindly ask for a more detailed explanation if possible. Assuming the object is fixed and the camera rotates around it to render the object from different viewpoints, what specific spatial information in the rendered images changes as the viewpoint shifts? Additionally, could you explain precisely what is meant by "reproject one of the views to another"? Thank you very much for your valuable time and assistance.
>
> >Q5. What does the performance preservation loss do in Eqn. (10)? Why would it be effective at a high level?
>
> The reason for introducing Eq. (10) is that, when training ControlNet with a limited number of images (in this case, rendered images from viewpoints close to the first viewpoint), the model is at high risk of forgetting the information it had previously learned. This results in a phenomenon known as catastrophic forgetting, which can significantly impair ControlNet's performance. To address this issue, Eq. (10) ensures that the parameters of ControlNet remain close to the values learned during its original training on a large dataset.
>
> >Q6. Writing and figure problems
>
> Thank you for your thorough review of our work and for highlighting areas in our writing and figures that could be improved. We sincerely apologize if these issues caused any difficulty in understanding or disrupted your reading experience. We deeply value your feedback and will carefully review the manuscript, including all figures, to address potential issues and ensure clarity and precision in the final version after incorporating feedback from all reviewers.
>
> >Q7. The authors do not show any viewpoint-varying results in video format, making it less convincing that TexTailor achieves a good view consistency.
>
> In addition to the four objects from the Objaverse dataset presented in the qualitative results of the paper, we will showcase qualitative comparisons for four additional objects from the same dataset in GIF format. Additionally, we will provide qualitative comparison results for clothed human meshes from RenderPeople[6], also presented in GIF format.
>
> The text prompts for RenderPeople's clothed human meshes were created based on the ground truth images provided by RenderPeople, which were designed by professional designers.

---

> ### Author Response · Authors · 2024-11-23
>
> #### 1.Objaverse Dataset
> |Text Prompt|Latent-Paint|Text2Tex|TEXTure|Paint-it|Ours|
> |:-----:|:-----:|:-----:|:-----:|:-----:|:-----:|
> |"a basketball"|[link](https://drive.google.com/file/d/1PUhh1CFVGfzvHqeNS7-VWlspWygV7M9i/view?usp=drive_link)|[link](https://drive.google.com/file/d/10-Vjeq2lBFIrApNeJ409yoJL3tbdL7M5/view?usp=drive_link)|[link](https://drive.google.com/file/d/1-zmxggvpEi17NlXk4rwqswcguWHToUnx/view?usp=drive_link)|[link](https://drive.google.com/file/d/1tGnNDX-1Z1AWRACQzloKAnW7EJrbkIgV/view?usp=drive_link)|[link](https://drive.google.com/file/d/1ChOOEJCHugwlXSTZMtRPVyR7MOBsXiow/view?usp=drive_link)|
> |"a cd player"|[link](https://drive.google.com/file/d/1fKhj-8aeGkVKjfb3FwtGUBkuf7FuKiM2/view?usp=sharing)|[link](https://drive.google.com/file/d/1VdXOQslsVru-x2x19tGYfKttT9rI7hit/view?usp=sharing)|[link](https://drive.google.com/file/d/1JE_LDcZm1-oBX4QJN-dIlmBCcNzJz2ne/view?usp=sharing)|[link](https://drive.google.com/file/d/1sRkv6iXBa7e9TVSMcV3hrlbZ8tZo0u_m/view?usp=sharing)|[link](https://drive.google.com/file/d/17k4XOBHya95ZsYwqRs_NONUff9Apw7r4/view?usp=sharing)|
> |"a desk chair"|[link](https://drive.google.com/file/d/1mm_0cvW2RsvktsLcy07JXAeS5CGVfENO/view?usp=drive_link)|[link](https://drive.google.com/file/d/1yJQkmI6THlNn3hoL-mW2O7D5CI_A6gq9/view?usp=drive_link)|[link](https://drive.google.com/file/d/1pRvTW1eEjVNk7rycoJRAJ8IdvPqHEj46/view?usp=drive_link)|[link](https://drive.google.com/file/d/1Jpx9vtBIV7ymD7zRHLBIPSNAxSItMjFe/view?usp=drive_link)|[link](https://drive.google.com/file/d/1ELVqtUhQbT5XmUcqko0h1pJjMGaicmAl/view?usp=drive_link)|
> |"a dumpster"|[link](https://drive.google.com/file/d/1HKaoGX_dUN0-TQXlmIAuCbHygUFUV3bD/view?usp=drive_link)|[link](https://drive.google.com/file/d/1f1m09_mCKa9_4KZOGKme8BwsAc_7lkJG/view?usp=drive_link)|[link](https://drive.google.com/file/d/18UZWI-sGhUqXNuFW0SdnIFFO_QNVoiNI/view?usp=drive_link)|[link](https://drive.google.com/file/d/1zN_VombixmEsueJfd6L5kP7zws5fzq0G/view?usp=drive_link)|[link](https://drive.google.com/file/d/1jtiXfphExSoW4wGiDGUBRTD1t4nGVdKi/view?usp=drive_link)|
> |"a hammer"|[link](https://drive.google.com/file/d/1RSq-vzcPfclttJToHObHypn7hs17VLEd/view?usp=sharing)|[link](https://drive.google.com/file/d/1Kw7aEhSdx3mCXdYkGKRzrGeobqaLJwAW/view?usp=sharing)|[link](https://drive.google.com/file/d/1aexhUd7Nl0Aey3Vg4I-XnoAdlzb3UZPJ/view?usp=drive_link)|[link](https://drive.google.com/file/d/1kxIuyKW3ROjT25KaohiRxpwVAwvQaVWC/view?usp=sharing)|[link](https://drive.google.com/file/d/1GqPePJEubcHmS5LOnMir5OmVIsHv8iTL/view?usp=drive_link)|
> |"a minivan"|[link](https://drive.google.com/file/d/1yEG9Ni4rrNaZ4kCvL4WRgrY5nU2CxfZS/view?usp=drive_link)|[link](https://drive.google.com/file/d/11nAvcQqOKF-fCaP5-ZAHBNTzD27Fujz2/view?usp=drive_link)|[link](https://drive.google.com/file/d/1sUQbNuGWEw9S85RozmnzII0ByIneFv3W/view?usp=drive_link)|[link](https://drive.google.com/file/d/1ku15fjR4ts_2KJm0Rl6OqGi5h9mP3hqi/view?usp=drive_link)|[link](https://drive.google.com/file/d/1EJcNOro3_wlw6spZ7TYFwtWfZ1Vx3dbL/view?usp=drive_link)|
> |"a money"|[front](https://drive.google.com/file/d/10rPPgZsWc0iotldevklujifOZMbiz-0X/view?usp=drive_link), [back](https://drive.google.com/file/d/1PaRrWMzpS073RU7bg8NKvnB2wVvnF1ZX/view?usp=drive_link)|[front](https://drive.google.com/file/d/1uNwXkh7O4UASFmm_BhDJJfX8MYWj_My6/view?usp=drive_link), [back](https://drive.google.com/file/d/10x-gy15J7OgJdymqFsalM_R-aF-XCJi4/view?usp=drive_link)|[front](https://drive.google.com/file/d/1rp4m1AdduMMOUyXMHUng5eq8jmQBsF1e/view?usp=drive_link), [back](https://drive.google.com/file/d/1mRWQlFv7YF4SeOrPSq4GrXkro6nveYzC/view?usp=drive_link)|[front](https://drive.google.com/file/d/1EFDwk-csQEiokhU09NiubX-yGC4NNXr7/view?usp=drive_link), [back](https://drive.google.com/file/d/1I_uEbvFRMn9HdfhjgDSNf7miW_fgTLQf/view?usp=drive_link)|[front](https://drive.google.com/file/d/1SRVIlpRovqU2o2d1XYQMkdHXsgd_TQxC/view?usp=drive_link), [back](https://drive.google.com/file/d/1Oc45Uj5M5ySbKikOYqvl6jA0KX9sEB6O/view?usp=drive_link)|
> |"a sushi"|[link](https://drive.google.com/file/d/15QWzVwVvJ-r7rFqUcVXGhnmNGRMdUClx/view?usp=drive_link)|[link](https://drive.google.com/file/d/1d7hxtTi8NFio4kAXeMJjIGyrIaOwmfmA/view?usp=drive_link)|[link](https://drive.google.com/file/d/1uZerJAnPSwV-bAw0e3JXwBpaPsL96XzK/view?usp=drive_link)|[link](https://drive.google.com/file/d/1VMeuBZoB-CuDDc0ekSCb5ezrXQLeVNgr/view?usp=drive_link)|[link](https://drive.google.com/file/d/1gSCl1HYbhxIK4ovEEs0OJVF1CgqIgHLg/view?usp=drive_link)|

---

> ### Author Response · Authors · 2024-11-23
>
> #### 2.RenderPeople Dataset
> |Text Prompt|Latent-Paint|Text2Tex|TEXTure|Paint-it|Ours|
> |:-----:|:-----:|:-----:|:-----:|:-----:|:-----:|
> |"A business woman wearing a white blouse with a ribbon detail, light beige pants, nude-tone heels, and neatly tied blonde hair"|[claudia](https://drive.google.com/file/d/1cugJNiHxkIxzZmfapOkUSUt1b5nmX-I4/view?usp=drive_link)|[claudia](https://drive.google.com/file/d/1tuUSCLRtZs8meKal50_kkfX0XkTDIlwU/view?usp=drive_link)|[claudia](https://drive.google.com/file/d/1HkO7HNKWkFh4u_xZfxRTgiyQV35ENyvS/view?usp=drive_link)|[claudia](https://drive.google.com/file/d/1cYYHgVoRn5YN-BQGbSXxtVa3ONWLBpIU/view?usp=drive_link)|[claudia](https://drive.google.com/file/d/1jpUILlxb39eSsAhZnk0kMBt2dTQ_iswl/view?usp=drive_link)|
> |"A man, wearing a white dress shirt, a black vest, black formal trousers, a black tie, a black belt, and dark formal shoes, with short neatly styled hair"|[eric](https://drive.google.com/file/d/1zu8JFHMlV2oHCu84je9z-hJn1_YUACl0/view?usp=drive_link)|[eric](https://drive.google.com/file/d/1ejcq1bB6djVUmmCqAxbqrDA2q_l_mOsK/view?usp=drive_link)|[eric](https://drive.google.com/file/d/1kQ4JW-Nu14p8qiLvR4RwvB90Iu_elRoI/view?usp=drive_link)|[eric](https://drive.google.com/file/d/1FItS8NhnffnzjoNPPU1or_t6ScWUveO2/view?usp=drive_link)|[eric](https://drive.google.com/file/d/1YibXGY7AicDrCaalaJUChKGRE5nQ6T7c/view?usp=drive_link)|
> |"A man, wearing a gray short-sleeve T-shirt, blue jeans, white sneakers, and short, dark brown hair styled neatly"|[manuel](https://drive.google.com/file/d/1yRmwmkkZ8c2VmC3fr2B92R2ztWT6BC0a/view?usp=sharing)|[manuel](https://drive.google.com/file/d/1jasYCugvfEaWtXMusmCFExluKTggHtKX/view?usp=sharing)|[manuel](https://drive.google.com/file/d/1kXzUleXzPhJkehtuoNUPhcj9YJzqYSRm/view?usp=sharing)|[manuel](https://drive.google.com/file/d/1ZYkWlhq2cxa-jyxaPeHbdvqTmjQE9-RP/view?usp=sharing)|[manuel](https://drive.google.com/file/d/1epeP2Ms2HC5AkW7uumw9f08H6dk-FVCc/view?usp=sharing)|
> |"A woman with medium-dark skin tone, wearing a black blazer, a black top, gray pants with a gray tied belt, black heels, and having neatly styled dark hair"|[carla](https://drive.google.com/file/d/1SN1Hft8lAbLU4HHTxB7k6e_WxoG-h-r-/view?usp=drive_link)|[carla](https://drive.google.com/file/d/1jJCNNpgXsB8xJKu6NnGkXicQ3ZDUOCRf/view?usp=drive_link)|[carla](https://drive.google.com/file/d/1OZ-uWrrX06M19CDgpukyCMkM0A5A4l78/view?usp=drive_link)|[carla](https://drive.google.com/file/d/1vc3EaAvNO18t8bUo8l_-KC5hhGTOzDTD/view?usp=sharing)|[carla](https://drive.google.com/file/d/1TlZ2wcokdUOz9Q-KIsSIShCiheelB1ox/view?usp=sharing)|
>
> ### Reference
> [1] Lugmayr, Andreas, et al. "Repaint: Inpainting using denoising diffusion probabilistic models." Proceedings of the IEEE/CVF conference on computer vision and pattern recognition. 2022.
>
> [2] Ho, Jonathan, Ajay Jain, and Pieter Abbeel. "Denoising diffusion probabilistic models." Advances in neural information processing systems 33 (2020): 6840-6851.
>
> [3] Song, Jiaming, Chenlin Meng, and Stefano Ermon. "Denoising diffusion implicit models." arXiv preprint arXiv:2010.02502 (2020).
>
> [4] xatlas, https://github.com/jpcy/xatlas
>
> [5] Laine, Samuli, et al. "Modular primitives for high-performance differentiable rendering." ACM Transactions on Graphics (ToG) 39.6 (2020): 1-14.
>
> [6] RenderPeople, https://renderpeople.com/free-3d-people/, 2023

---

> ### Comment · Reviewer_ujQC · 2024-11-24
>
> Thanks for your responses! Most of my concerns are revolved. I'd like to increase my rating from 3 to 5.
>
> I still have a concern about Q4. I'd like to clarify my point:
> - LPIPS, although as a "corase" perceptual loss, still shows high error for the pixels that are not aligned well. For example, if you rotate your image by 30 degrees, you will see the LPIPS between the original image and the rotated one becomes obvious.
> - The authors mention that they measure the view-consistency by computing LPIPS between rendered images across different viewpoints (L400-L402). In that sense, will LPIPS becomes unreliable since it is sensitive to this "spatial shift" in the example, although they contain high-level information?
> - To better compute the view-consistency, a possible way could be that we project the image from one viewpoint $v_1$ to another one $v_2$, and then compute the difference (you could use LPIPS) between the rendered image at $v_2$ and the "reprojected" image from $v_1$.

---

> > ### Author Response · Authors · 2024-11-24
> >
> > Based on my understanding, the current question seems to address the overlapping texture regions between adjacent viewpoints. For example, when rendering from two viewpoints, $v_1$  and $v_2$, a portion of the mesh surface visible from $v_1$ may appear rotated or differently positioned when viewed from $v_2$. This overlap may affect the LPIPS metric and raises concerns about its reliability as a measure of view consistency.
> >
> > However, we believe LPIPS remains a valuable metric for measuring view consistency for the following reasons:
> >
> > 1. All models compared in the paper were rendered from the same set of viewpoints. As such, any increase in LPIPS due to positional differences in the overlapping regions is equally reflected across all models. Therefore, the differences in LPIPS scores presented in the paper are not a result of positional discrepancies but rather the perceptual differences between the rendered images (i.e., how naturally texture consistency is maintained across viewpoints).
> >
> > 2. There are prior works [1] that have used LPIPS as a metric for measuring view consistency, which lends credibility to its application in this context.
> >
> > 3. To address the overlap between $v_1$ and $v_2$  through reprojection, one simple approach involves using SIFT to extract feature points and estimate a Homography Matrix via feature matching. However, since this method relies on estimation, it may introduce inaccuracies or distortions, which could undermine the reliability of the results.
> >
> > ### Reference
> > [1] Hong, Susung, Donghoon Ahn, and Seungryong Kim. "Debiasing scores and prompts of 2d diffusion for view-consistent text-to-3d generation." Advances in Neural Information Processing Systems 36 (2023): 11970-11987.

---

> > > ### Author Response · Authors · 2024-11-24
> > >
> > > Additionally, I would like to take this opportunity to clarify the contributions of my paper, as I believe they may not have been explicitly highlighted in the manuscript.
> > >
> > > I believe the contributions of this paper are as follows:
> > > 1. We extend the resampling technique, previously used only in 2D image inpainting, to the field of 3D texture synthesis by applying it to the DDIM non-Markovian process.
> > > 2. Without relying on an external dataset of 3D meshes, textures, or text, we propose a novel approach that fine-tunes the model using only a few images that accurately represent the texture of a specific object within the distribution learned by the existing depth-aware T2I model. Even with just a few images inferred by the original model, this approach effectively compensates for textures that are difficult to generate at certain angles and demonstrates the ability to maintain texture consistency across various viewpoints
> > > 3. To address the catastrophic forgetting phenomenon—where a depth-aware T2I model forgets information originally learned from a large-scale dataset when fine-tuned on a smaller one—we propose the performance preservation loss to mitigate this issue and maintain the model's performance.
> > > 4. To eliminate the need for manually configuring optimal camera positions based on object geometry—a process requiring significant time and effort—we introduce an adaptive method that adjusts camera positions dynamically based on the extent of texture coverage at each viewpoint.
> > >
> > > we believe that the other contributions, particularly the ability to generate consistent images across various angles, can be highly applicable not only to the texture synthesis field but also to other 3D domains requiring such consistency. This makes our work a valuable contribution to the ICLR community and a potential foundation for future research.

---

> > > > ### Author Response · Authors · 2024-11-27
> > > >
> > > > Thank you for your detailed feedback and for raising your concern about Q4. I hope the response we provided sufficiently addressed your question regarding the use of LPIPS for measuring view-consistency and the potential alternative of reprojecting viewpoints. If there are any remaining points of clarification needed or further concerns, please do not hesitate to let us know. Your insights are greatly valued, and we want to ensure we have addressed your feedback thoroughly.

---

> ### Comment · Reviewer_ujQC · 2024-11-27
>
> Thanks for providing more explanations about the metrics. It looks like LPIPS has been there for a while in this line of work but nobody finds its flaws as a metric in this task.
>
> I'll increase my rating. Please incorporate the discussions in the revision.
>
> Cheers.

---

### Official Review · Reviewer_3JDj · 2024-10-31

**Soundness:** 3
**Presentation:** 3
**Contribution:** 2
**Rating:** 6
**Confidence:** 4

**Summary:**

This paper introduces a method for text-driven 3D object texturing. Previous work on this important topic fail in some areas, according to the paper, including: Consistency and the gradual change in textures assigned to the object. The paper aims to solve these issues, by introducing 2 ideas: First, the model leverages a resampling scheme for better integration of previously generated texture during the diffusion process, and second, the model fine-tunes a depth-aware diffusion model with these resampling textures. With these contributions, the method is said to achieve higher quality and consistency than previous work, measured on a set of datasets and perceptual metrics.

**Strengths:**

- This paper adequately identifies problems in previous methods for text-driven object texturing, including lack of texture consistency and graduality in texture changes. The origin of this problems are identified as being caused by insufficient integration, predefinition of camera positions, and autorregresion. The paper introduces changes to these methods, to enhance their quality and consistency. This is an important line of research, as these works are becoming more prevalent in the literature and industrial applications.
- This paper tackles an important and salient problem in the literature.
- The results shown in the paper indeed suggest that the proposed method provides less gradual changes in texture properties. In objects with different parts, TexTailor shows superior performance in assigning different texture to different parts, than previous work do.
- The proposed method is sound, and the ideas proposed here are very well suited for the task the paper is aiming to solve. In this sense, the paper is correct as far as I am familiar with the literature and the problems in 3D content generation.
- This paper is well written and easy to follow. The problems identified in previous work are clearly stated, and the ideas to solve them are easy to understand and very well explained.
- Code is provided as a supplementary material, which should greatly enhance reproducibility.

**Weaknesses:**

- While sound, the ideas introduced in this work are somewhat limited in scope and the paper fails to be compelling that they are particularly effective. In this sense, I am not convinced about the extent upon which these contributions will be impactful in the literature. Furthermore, the resampling scheme introduced in this paper is not new, as it is borrowed from previous work. Therefore, the ideas introduced here are not particularly novel nor signficant.
- Insufficient results are shown on the paper. It is hard to understand the capabilities of the model with the amount of results shown here. In particular, only four results are shown in the comparisons on Objaverse, and these results are not particularly compelling (for example, in the hammer, the method assigns a metal texture to the handle and a wooden texture to the head, which is not correct and arguably a worse result than TEXTure). Only two results are shown in ShapeNet car, and the ablation study is shown exclusively on a single object. Significant more results should be provided to convince the reader that the method is more effective than previous work.
- I am unconvinced about the metrics used in this paper. While standard for 3D object texturing work, LPIPS and FID do not adequately measure text-to-image alignment. CLIP-based metrics should be used in conjunction to the ones shown in this paper, to be more informative about how well this model is generating results aligned with the prompts. While visually more consistent than previous work, this model seems to struggle more than previous methods (particularly TEXTure and Text2Text) in asigning the correct texture to each part of the object. This is not something that LPIPS and FID can measure correctly.
- A user study should be provided for better comparisons between methods, across a bunch of dimensions, including alignment, realism, quality, consistency, etc.
- The quantitative results are not particularly convincing. The ablation study does not show significant improvements across the metrics used, particularly LPIPS, and without standard deviations of the errors it is hard to understand whether the improvements are actually statistically significant. Therefore, the ablation fails to convince that the proposed contributions are actually valuable and effective.
- No results are shown on 3D human avatar texturing, which is a very closely related and relevant line of work.
- Related to the previous point, the analysis of the related work is lacking on a set of areas. The most relevant is the work on 3D human texturing. Relevant work include: SMPLitex: A Generative Model and Dataset for 3D Human Texture Estimation from Single Image (BMVC 2023), UVMap-ID: A Controllable and Personalized UV Map Generative Model (ACMMM 2024), TexDreamer: Towards Zero-Shot High-Fidelity 3D Human Texture Generation (ECCV 2024), etc. Besides, 2D texturing methods with generative models should also be included as part of the related work.
- Limitations are not adequately addressed or discussed. From the paper, it seems like the only limitation of the proposed method is its computational cost. However, the results shown in the paper indicate that the method is by no means perfect and it struggles with consistently assigning the appropriate texture to different parts of the object, among other limitations. These should be mentioned more explicitly.
- Contributions are very overstated. Sentences like "... demonstrate the superior performance of TexTailor in .... " or " ... surpases SOTA texture synthesis methods driven by language cues" should be empirically demonstrated or removed altogether.
- The paper suggests some reasons why previous methods fail (autorregressive inference, integration of previous information, fixed camera positions, etc), but it fails to provide adequate evidence that these actually limiting factors.

**Questions:**

- Can the authors provide a user study that measures consistency, alignment, quality, and realism? This should provide a better idea on the quality of the results on the actual goals that the paper aims to achieve.
- I believe that computational costs should be compared more explictly with previous work, so as to better understand the quality/cost pareto frontier in this line of work.
- I suggest the paper should provide a CLIP-guided text-image alignment metric.
- I suggest the paper should provide a more upfront discussion of its limitations.
- The paper should include a detailed analysis of 2D texturing models.
- The paper should include a detailed analysis of text-to-avatar models, as well as quantitative and qualitative comparisons.
- How does the model behave with non-diffuse objects? Very few glossy, metallic, or translucent objects are shown.
- The paper should include many more results, at least in the supplementary material.
- Results should include standard deviations to better understand the differences between methods in terms of LPIPS, FLIP, etc.

---

> ### Author Response · Authors · 2024-11-24
>
> Thank you for taking the time to review our work. We greatly appreciate your recognition of the key aspects of our proposed methods and your positive feedback on the experiments. We will revise the manuscript to incorporate your suggestions.
>
> > Q1. While sound, the ideas introduced in this work are somewhat limited in scope and the paper fails to be compelling that they are particularly effective. In this sense, I am not convinced about the extent upon which these contributions will be impactful in the literature. Furthermore, the resampling scheme introduced in this paper is not new, as it is borrowed from previous work. Therefore, the ideas introduced here are not particularly novel nor signficant.
>
> I believe the contributions of this paper are as follows:
>
> 1.	We extend the resampling technique, previously used only in 2D image inpainting, to the field of 3D texture synthesis by applying it to the DDIM non-Markovian process.
>
> 2.	Without relying on an external dataset of 3D meshes, textures, or text, we propose a novel approach that fine-tunes the model using only a few images that accurately represent the texture of a specific object within the distribution learned by the existing depth-aware T2I model. Even with just a few images inferred by the original model, this approach effectively compensates for textures that are difficult to generate at certain angles and demonstrates the ability to maintain texture consistency across various viewpoints
>
> 3.	To address the catastrophic forgetting phenomenon—where a depth-aware T2I model forgets information originally learned from a large-scale dataset when fine-tuned on a smaller one—we propose the performance preservation loss to mitigate this issue and maintain the model's performance.
>
> 4.	To eliminate the need for manually configuring optimal camera positions based on object geometry—a process requiring significant time and effort—we introduce an adaptive method that adjusts camera positions dynamically based on the extent of texture coverage at each viewpoint.
>
> As you mentioned, point 1 may seem less novel as it recombines existing methods. However, we believe that the other contributions, particularly the ability to generate consistent images across various angles, can be highly applicable not only to the texture synthesis field but also to other 3D domains requiring such consistency. This makes our work a valuable contribution to the ICLR community and a potential foundation for future research.
>
> >Q2. Insufficient results are shown on the paper. It is hard to understand the capabilities of the model with the amount of results shown here. In particular, only four results are shown in the comparisons on Objaverse, and these results are not particularly compelling (for example, in the hammer, the method assigns a metal texture to the handle and a wooden texture to the head, which is not correct and arguably a worse result than TEXTure). Only two results are shown in ShapeNet car, and the ablation study is shown exclusively on a single object. Significant more results should be provided to convince the reader that the method is more effective than previous work.
>
> To further demonstrate the effectiveness of our proposed methodology, we will showcase qualitative comparisons for four additional objects from the Objaverse dataset, beyond the four objects already presented in the qualitative results of the paper, in GIF format.
>
> Additionally, to demonstrate that the effectiveness of each method shown in Fig. 7 of the paper is not limited to "a muffin" object, but is also applicable to other types of objects, we will present ablation studies on additional objects as well.

---

> > ### Author Response · Authors · 2024-11-24
> >
> > #### 1.Objaverse Dataset
> > |Text Prompt|Latent-Paint|Text2Tex|TEXTure|Paint-it|Ours|
> > |:-----:|:-----:|:-----:|:-----:|:-----:|:-----:|
> > |"a basketball"|[link](https://drive.google.com/file/d/1PUhh1CFVGfzvHqeNS7-VWlspWygV7M9i/view?usp=drive_link)|[link](https://drive.google.com/file/d/10-Vjeq2lBFIrApNeJ409yoJL3tbdL7M5/view?usp=drive_link)|[link](https://drive.google.com/file/d/1-zmxggvpEi17NlXk4rwqswcguWHToUnx/view?usp=drive_link)|[link](https://drive.google.com/file/d/1tGnNDX-1Z1AWRACQzloKAnW7EJrbkIgV/view?usp=drive_link)|[link](https://drive.google.com/file/d/1ChOOEJCHugwlXSTZMtRPVyR7MOBsXiow/view?usp=drive_link)|
> > |"a cd player"|[link](https://drive.google.com/file/d/1fKhj-8aeGkVKjfb3FwtGUBkuf7FuKiM2/view?usp=sharing)|[link](https://drive.google.com/file/d/1VdXOQslsVru-x2x19tGYfKttT9rI7hit/view?usp=sharing)|[link](https://drive.google.com/file/d/1JE_LDcZm1-oBX4QJN-dIlmBCcNzJz2ne/view?usp=sharing)|[link](https://drive.google.com/file/d/1sRkv6iXBa7e9TVSMcV3hrlbZ8tZo0u_m/view?usp=sharing)|[link](https://drive.google.com/file/d/17k4XOBHya95ZsYwqRs_NONUff9Apw7r4/view?usp=sharing)|
> > |"a desk chair"|[link](https://drive.google.com/file/d/1mm_0cvW2RsvktsLcy07JXAeS5CGVfENO/view?usp=drive_link)|[link](https://drive.google.com/file/d/1yJQkmI6THlNn3hoL-mW2O7D5CI_A6gq9/view?usp=drive_link)|[link](https://drive.google.com/file/d/1pRvTW1eEjVNk7rycoJRAJ8IdvPqHEj46/view?usp=drive_link)|[link](https://drive.google.com/file/d/1Jpx9vtBIV7ymD7zRHLBIPSNAxSItMjFe/view?usp=drive_link)|[link](https://drive.google.com/file/d/1ELVqtUhQbT5XmUcqko0h1pJjMGaicmAl/view?usp=drive_link)|
> > |"a dumpster"|[link](https://drive.google.com/file/d/1HKaoGX_dUN0-TQXlmIAuCbHygUFUV3bD/view?usp=drive_link)|[link](https://drive.google.com/file/d/1f1m09_mCKa9_4KZOGKme8BwsAc_7lkJG/view?usp=drive_link)|[link](https://drive.google.com/file/d/18UZWI-sGhUqXNuFW0SdnIFFO_QNVoiNI/view?usp=drive_link)|[link](https://drive.google.com/file/d/1zN_VombixmEsueJfd6L5kP7zws5fzq0G/view?usp=drive_link)|[link](https://drive.google.com/file/d/1jtiXfphExSoW4wGiDGUBRTD1t4nGVdKi/view?usp=drive_link)|
> > |"a hammer"|[link](https://drive.google.com/file/d/1RSq-vzcPfclttJToHObHypn7hs17VLEd/view?usp=sharing)|[link](https://drive.google.com/file/d/1Kw7aEhSdx3mCXdYkGKRzrGeobqaLJwAW/view?usp=sharing)|[link](https://drive.google.com/file/d/1aexhUd7Nl0Aey3Vg4I-XnoAdlzb3UZPJ/view?usp=drive_link)|[link](https://drive.google.com/file/d/1kxIuyKW3ROjT25KaohiRxpwVAwvQaVWC/view?usp=sharing)|[link](https://drive.google.com/file/d/1GqPePJEubcHmS5LOnMir5OmVIsHv8iTL/view?usp=drive_link)|
> > |"a minivan"|[link](https://drive.google.com/file/d/1yEG9Ni4rrNaZ4kCvL4WRgrY5nU2CxfZS/view?usp=drive_link)|[link](https://drive.google.com/file/d/11nAvcQqOKF-fCaP5-ZAHBNTzD27Fujz2/view?usp=drive_link)|[link](https://drive.google.com/file/d/1sUQbNuGWEw9S85RozmnzII0ByIneFv3W/view?usp=drive_link)|[link](https://drive.google.com/file/d/1ku15fjR4ts_2KJm0Rl6OqGi5h9mP3hqi/view?usp=drive_link)|[link](https://drive.google.com/file/d/1EJcNOro3_wlw6spZ7TYFwtWfZ1Vx3dbL/view?usp=drive_link)|
> > |"a money"|[front](https://drive.google.com/file/d/10rPPgZsWc0iotldevklujifOZMbiz-0X/view?usp=drive_link), [back](https://drive.google.com/file/d/1PaRrWMzpS073RU7bg8NKvnB2wVvnF1ZX/view?usp=drive_link)|[front](https://drive.google.com/file/d/1uNwXkh7O4UASFmm_BhDJJfX8MYWj_My6/view?usp=drive_link), [back](https://drive.google.com/file/d/10x-gy15J7OgJdymqFsalM_R-aF-XCJi4/view?usp=drive_link)|[front](https://drive.google.com/file/d/1rp4m1AdduMMOUyXMHUng5eq8jmQBsF1e/view?usp=drive_link), [back](https://drive.google.com/file/d/1mRWQlFv7YF4SeOrPSq4GrXkro6nveYzC/view?usp=drive_link)|[front](https://drive.google.com/file/d/1EFDwk-csQEiokhU09NiubX-yGC4NNXr7/view?usp=drive_link), [back](https://drive.google.com/file/d/1I_uEbvFRMn9HdfhjgDSNf7miW_fgTLQf/view?usp=drive_link)|[front](https://drive.google.com/file/d/1SRVIlpRovqU2o2d1XYQMkdHXsgd_TQxC/view?usp=drive_link), [back](https://drive.google.com/file/d/1Oc45Uj5M5ySbKikOYqvl6jA0KX9sEB6O/view?usp=drive_link)|
> > |"a sushi"|[link](https://drive.google.com/file/d/15QWzVwVvJ-r7rFqUcVXGhnmNGRMdUClx/view?usp=drive_link)|[link](https://drive.google.com/file/d/1d7hxtTi8NFio4kAXeMJjIGyrIaOwmfmA/view?usp=drive_link)|[link](https://drive.google.com/file/d/1uZerJAnPSwV-bAw0e3JXwBpaPsL96XzK/view?usp=drive_link)|[link](https://drive.google.com/file/d/1VMeuBZoB-CuDDc0ekSCb5ezrXQLeVNgr/view?usp=drive_link)|[link](https://drive.google.com/file/d/1gSCl1HYbhxIK4ovEEs0OJVF1CgqIgHLg/view?usp=drive_link)|

---

> ### Author Response · Authors · 2024-11-24
>
> #### 2. Ablation study
> |Figure|
> |:-----:|
> |[link](https://drive.google.com/file/d/1I8zhpnybAV0g-dka4Ct2Z028jFyOHD4C/view?usp=drive_link)|
>
> ##### (1) Effects of resampling.
>
> When observing the texture of "a basket," the transition from $v_0$ to $v_1$ reveals a change in the texture of the inside of the basket in the left part. This occurs in the baseline method because the previously synthesized texture visible from the current viewpoint is only merged once per timestep during the diffusion process. In contrast, our methodology improves this by utilizing the resampling strategy proposed in the Repaint[6] paper, which allows the texture to be merged multiple times per timestep, resulting in a more consistent outcome.
>
> ##### (2) Effects of training with resampled texture.
>
> When observing the object "a briefcase," applying the Resampling method generates consistent textures for viewpoints that are close to the initial viewpoint $v_0$, such as the nearby viewpoint $v_5$. However, as the viewpoint moves further away from $v_0$, variations in texture properties begin to appear, and at the opposite viewpoint $v_9$, the texture properties have completely changed. In contrast, when the depth-aware T2I model is fine-tuned using five resampled texture images from viewpoints adjacent to $v_0$, the model fits to the distribution of the resampled textures, leading to noticeable improvements in consistency.
>
>
> ##### (3) Effects of adaptive view refinement.
> When observing the object "a cappuccino," transitioning from $v_{11}$ to $v_{12}$ on the left part reveals an issue where the bottom of the cup is generated incorrectly. Since there is no texture information from the previous viewpoint, the model generates the top of the cappuccino instead of the bottom. However, by using the adaptive view refinement technique on the right part, an intermediate viewpoint ($v_{13}$) is automatically added. This not only provides the texture information from $v_{11}$ to guide a more natural texture synthesis but also eliminates the need for the tedious process of manually configuring optimal camera positions.
>
> > Q3. I am unconvinced about the metrics used in this paper. While standard for 3D object texturing work, LPIPS and FID do not adequately measure text-to-image alignment. CLIP-based metrics should be used in conjunction to the ones shown in this paper, to be more informative about how well this model is generating results aligned with the prompts. While visually more consistent than previous work, this model seems to struggle more than previous methods (particularly TEXTure and Text2Text) in asigning the correct texture to each part of the object. This is not something that LPIPS and FID can measure correctly. I suggest the paper should provide a CLIP-guided text-image alignment metric.
>
> As per your suggestion, we calculated the average cosine similarity between the prompt and image CLIP[1] embeddings for the four models compared in this paper.
>
> |Latent-Paint|Text2Tex|TEXTure|Paint-it|Ours|
> |:-----:|:-----:|:-----:|:-----:|:-----:|
> |0.2334|0.2295|0.2342|0.2315|0.2323|
>
> As shown in the results, while TexTailor does not achieve the highest value compared to other models, the differences between the values across models are not significant enough to be considered meaningful. Therefore, we plan to address this point when incorporating all reviewers' feedback in the final version. Thank you for your valuable feedback and thoughtful suggestion.
>
> > Q4. A user study should be provided for better comparisons between methods, across a bunch of dimensions, including alignment, realism, quality, consistency, etc. Can the authors provide a user study that measures consistency, alignment, quality, and realism? This should provide a better idea on the quality of the results on the actual goals that the paper aims to achieve.
>
> Thank you for suggesting a user study to evaluate consistency, alignment, quality, and realism. We agree that such a study would provide valuable insights into the quality of our results and how well they align with the goals of the paper. However, due to resource and time constraints, we were unable to recruit a sufficient number of users to conduct a statistically meaningful user study during this submission cycle. That said, we acknowledge the importance of this evaluation and plan to include a comprehensive user study in future work to further substantiate our findings.

---

> > ### Author Response · Authors · 2024-11-24
> >
> > > Q5. The quantitative results are not particularly convincing. The ablation study does not show significant improvements across the metrics used, particularly LPIPS, and without standard deviations of the errors it is hard to understand whether the improvements are actually statistically significant. Therefore, the ablation fails to convince that the proposed contributions are actually valuable and effective. and The paper suggests some reasons why previous methods fail (autorregressive inference, integration of previous information, fixed camera positions, etc), but it fails to provide adequate evidence that these actually limiting factors. Results should include standard deviations to better understand the differences between methods in terms of LPIPS, FLIP, etc.
> >
> > I agree with your point. Not including the standard deviations of the errors in quantitative comparisons can indeed reduce the reliability of the experiments. However, we regret that due to time and resource constraints during the rebuttal period, we were unable to repeat all experiments multiple times, and we sincerely apologize for this limitation.
> >
> > We greatly value your feedback and will make an effort to include standard deviations in future experiments to enhance reliability. Additionally, we kindly ask for your understanding, as the models compared in this paper also did not include standard deviations in their results. Thank you for your thoughtful feedback.
> >
> > > Q6. No results are shown on 3D human avatar texturing, which is a very closely related and relevant line of work. Related to the previous point, the analysis of the related work is lacking on a set of areas. The most relevant is the work on 3D human texturing. Relevant work include: SMPLitex: A Generative Model and Dataset for 3D Human Texture Estimation from Single Image (BMVC 2023), UVMap-ID: A Controllable and Personalized UV Map Generative Model (ACMMM 2024), TexDreamer: Towards Zero-Shot High-Fidelity 3D Human Texture Generation (ECCV 2024), etc. Besides, 2D texturing methods with generative models should also be included as part of the related work. The paper should include a detailed analysis of text-to-avatar models, as well as quantitative and qualitative comparisons.
> >
> > I completely agree with your point. 3D human avatar texturing is a highly active research area and holds significant importance in various applications. However, unfortunately, the methods you mentioned—SMPLitex[2], UVMap-ID[3], and TexDreamer[4]—are aimed at image-guided texture synthesis rather than text-guided texture synthesis, making direct comparisons challenging.
> >
> > Additionally, we were unable to find an appropriate dataset combining 3D clothed human meshes with corresponding text descriptions for evaluation. As such, we conducted qualitative comparisons on four clothed human meshes provided by RenderPeople[5], using the four models currently compared in the paper. We kindly ask for your understanding regarding this limitation.
> >
> > The text prompts for RenderPeople's clothed human meshes were created based on the ground truth images provided by RenderPeople, which were designed by professional designers.

---

> > > ### Author Response · Authors · 2024-11-24
> > >
> > > #### RenderPeople Dataset
> > > |Text Prompt|Latent-Paint|Text2Tex|TEXTure|Paint-it|Ours|
> > > |:-----:|:-----:|:-----:|:-----:|:-----:|:-----:|
> > > |"A business woman wearing a white blouse with a ribbon detail, light beige pants, nude-tone heels, and neatly tied blonde hair"|[claudia](https://drive.google.com/file/d/1cugJNiHxkIxzZmfapOkUSUt1b5nmX-I4/view?usp=drive_link)|[claudia](https://drive.google.com/file/d/1tuUSCLRtZs8meKal50_kkfX0XkTDIlwU/view?usp=drive_link)|[claudia](https://drive.google.com/file/d/1HkO7HNKWkFh4u_xZfxRTgiyQV35ENyvS/view?usp=drive_link)|[claudia](https://drive.google.com/file/d/1cYYHgVoRn5YN-BQGbSXxtVa3ONWLBpIU/view?usp=drive_link)|[claudia](https://drive.google.com/file/d/1jpUILlxb39eSsAhZnk0kMBt2dTQ_iswl/view?usp=drive_link)|
> > > |"A man, wearing a white dress shirt, a black vest, black formal trousers, a black tie, a black belt, and dark formal shoes, with short neatly styled hair"|[eric](https://drive.google.com/file/d/1zu8JFHMlV2oHCu84je9z-hJn1_YUACl0/view?usp=drive_link)|[eric](https://drive.google.com/file/d/1ejcq1bB6djVUmmCqAxbqrDA2q_l_mOsK/view?usp=drive_link)|[eric](https://drive.google.com/file/d/1kQ4JW-Nu14p8qiLvR4RwvB90Iu_elRoI/view?usp=drive_link)|[eric](https://drive.google.com/file/d/1FItS8NhnffnzjoNPPU1or_t6ScWUveO2/view?usp=drive_link)|[eric](https://drive.google.com/file/d/1YibXGY7AicDrCaalaJUChKGRE5nQ6T7c/view?usp=drive_link)|
> > > |"A man, wearing a gray short-sleeve T-shirt, blue jeans, white sneakers, and short, dark brown hair styled neatly"|[manuel](https://drive.google.com/file/d/1yRmwmkkZ8c2VmC3fr2B92R2ztWT6BC0a/view?usp=sharing)|[manuel](https://drive.google.com/file/d/1jasYCugvfEaWtXMusmCFExluKTggHtKX/view?usp=sharing)|[manuel](https://drive.google.com/file/d/1kXzUleXzPhJkehtuoNUPhcj9YJzqYSRm/view?usp=sharing)|[manuel](https://drive.google.com/file/d/1ZYkWlhq2cxa-jyxaPeHbdvqTmjQE9-RP/view?usp=sharing)|[manuel](https://drive.google.com/file/d/1epeP2Ms2HC5AkW7uumw9f08H6dk-FVCc/view?usp=sharing)|
> > > |"A woman with medium-dark skin tone, wearing a black blazer, a black top, gray pants with a gray tied belt, black heels, and having neatly styled dark hair"|[carla](https://drive.google.com/file/d/1SN1Hft8lAbLU4HHTxB7k6e_WxoG-h-r-/view?usp=drive_link)|[carla](https://drive.google.com/file/d/1jJCNNpgXsB8xJKu6NnGkXicQ3ZDUOCRf/view?usp=drive_link)|[carla](https://drive.google.com/file/d/1OZ-uWrrX06M19CDgpukyCMkM0A5A4l78/view?usp=drive_link)|[carla](https://drive.google.com/file/d/1vc3EaAvNO18t8bUo8l_-KC5hhGTOzDTD/view?usp=sharing)|[carla](https://drive.google.com/file/d/1TlZ2wcokdUOz9Q-KIsSIShCiheelB1ox/view?usp=sharing)|
> > >
> > > >Q7. How does the model behave with non-diffuse objects? Very few glossy, metallic, or translucent objects are shown.
> > >
> > > We will provide two GIF results for each type of object you mentioned, showcasing the outcomes.
> > >
> > > #### 1. glossy objects
> > > |Text Prompt|"a bowl"|"a cup"|
> > > |:-----:|:-----:|:-----:|
> > > |Link|[link](https://drive.google.com/file/d/1qyHWpLgb26THiM0IHUhnJDa24fanzpnH/view?usp=drive_link)|[link](https://drive.google.com/file/d/1jkEb-Jo0uFfTsn4Q5FX7twzpVdMVJDnV/view?usp=drive_link)|
> > >
> > > #### 2. metallic objects
> > > |Text Prompt|"a faucet"|"a aerosel can"|
> > > |:-----:|:-----:|:-----:|
> > > |Link|[link](https://drive.google.com/file/d/1_JN5iot_k8loDKTHfAZbbn5tyznbtYdB/view?usp=drive_link)|[link](https://drive.google.com/file/d/1lWl5BTPW9vFro78TicnukQBcJDa10UOK/view?usp=drive_link)|
> > >
> > > #### 3. translucent objects
> > > |Text Prompt|"a lightbulb"|"a candle"|
> > > |:-----:|:-----:|:-----:|
> > > |Link|[link](https://drive.google.com/file/d/13E2HoLO1URUeeLCaM2NzPAxBYLK-S00e/view?usp=sharing)|[link](https://drive.google.com/file/d/1efGXb6KB0hHrfKuosUYukawbNckXCeax/view?usp=drive_link)|
> > >
> > > > Q9. The paper should include a detailed analysis of 2D texturing models.
> > >
> > > Are you referring to the four texture models mentioned in our paper when you say "2D texturing models"? If so, does the "detailed analysis" refer to the qualitative results presented in the paper?
> > >
> > > > Q10. Contributions are very overstated. Sentences like "... demonstrate the superior performance of TexTailor in .... " or " ... surpases SOTA texture synthesis methods driven by language cues" should be empirically demonstrated or removed altogether.
> > >
> > > Thank you for pointing this out. We understand your concern regarding the overstatement of contributions. Once we have gathered all the reviewers' feedback, we will revise the manuscript comprehensively for the final version, ensuring that all claims are either empirically demonstrated or appropriately adjusted. We sincerely appreciate your constructive input.

---

> > > > ### Author Response · Authors · 2024-11-24
> > > >
> > > > > Q11. I believe that computational costs should be compared more explictly with previous work, so as to better understand the quality/cost pareto frontier in this line of work.
> > > >
> > > > |Method|Latent-Paint|Text2Tex|TEXTure|Paint-it|Ours|
> > > > |:-----:|:-----:|:-----:|:-----:|:-----:|:-----:|
> > > > |Runtime(minutes)|46|24|2|25|90|
> > > > |GPU memory usage(MB)|10670|12077|12228|30905|22478|
> > > >
> > > > As shown, TexTailor has the longest runtime among the methods, which we acknowledge as one of the key limitations discussed in the paper. To overcome this limitation, accelerating the fine-tuning process is a primary focus for our future work.
> > > >
> > > > > Q12. Limitations are not adequately addressed or discussed. From the paper, it seems like the only limitation of the proposed method is its computational cost. However, the results shown in the paper indicate that the method is by no means perfect and it struggles with consistently assigning the appropriate texture to different parts of the object, among other limitations. These should be mentioned more explicitly. I suggest the paper should provide a more upfront discussion of its limitations.
> > > >
> > > > Beyond computational cost, we identify the following limitations of TexTailor:
> > > >
> > > > (1) Dependence on the Quality of the Five Training Images: The overall texture quality heavily relies on the quality of the five images used to fine-tune the depth-aware T2I diffusion model. Even with the resampling method and the use of viewpoints close to the initial one, certain angles of the object may still produce textures with inconsistent properties or images misaligned with the depth condition. Fine-tuning the model using such suboptimal images can potentially degrade the texture quality instead of improving it.
> > > >
> > > > (2) Repetitive Patterns Across Views: Patterns frequently observed in viewpoints close to the initial one tend to repeat throughout the object. For instance, in the case of an alarm clock, the clock hands learned from the initial viewpoint may appear as texture patterns on the sides or even the front of the clock, resulting in repetitive and unrealistic textures.
> > > >
> > > >
> > > > ### Reference
> > > > [1] Radford, Alec, et al. "Learning transferable visual models from natural language supervision." International conference on machine learning. PMLR, 2021.
> > > >
> > > > [2] Casas, Dan, and Marc Comino-Trinidad. "Smplitex: A generative model and dataset for 3d human texture estimation from single image." arXiv preprint arXiv:2309.01855 (2023).
> > > >
> > > > [3] Wang, Weijie, et al. "UVMap-ID: A Controllable and Personalized UV Map Generative Model." Proceedings of the 32nd ACM International Conference on Multimedia. 2024.
> > > >
> > > > [4] Liu, Yufei, et al. "TexDreamer: Towards Zero-Shot High-Fidelity 3D Human Texture Generation." arXiv preprint arXiv:2403.12906 (2024).
> > > >
> > > > [5] RenderPeople, https://renderpeople.com/free-3d-people/, 2023
> > > >
> > > > [6] Lugmayr, Andreas, et al. "Repaint: Inpainting using denoising diffusion probabilistic models." Proceedings of the IEEE/CVF conference on computer vision and pattern recognition. 2022.

---

> ### Comment · Reviewer_3JDj · 2024-11-25
> **Response to rebuttal**
>
> I thank the authors for an oustandingly thorough rebuttal to my and other reviewers' concerns. While I remain unconvinced about the scope of the contributions and the lack of a user study, many of my original concerns have been adressed by the authors and I will therefore increase my rating accordingly.

---

### Official Review · Reviewer_uECh · 2024-11-02

**Soundness:** 3
**Presentation:** 3
**Contribution:** 2
**Rating:** 6
**Confidence:** 4

**Summary:**

This paper proposes TexTailor, a method for text-to-texture synthesis utilizing an inpainting approach to achieve view-consistent textures. To address common challenges in texture generation, TexTailor introduces a resampling scheme and fine-tuning to maintain texture consistency across viewpoints. Furthermore, it employs adaptive viewpoint refinement for efficient viewpoint sampling.

**Strengths:**

1. The paper proposes TexTailor to address view-consistent texture synthesis by combining inpainting with resampling and fine-tuning.
2. Method and results are presented clearly and logically, making the paper easy to follow.

**Weaknesses:**

1. While effective, the approach primarily combines existing techniques, with limited emphasis on novel contributions. The paper could be strengthened by enhancing the resampling scheme or accelerating the fine-tuning phase.

**Questions:**

1. Could the authors clarify the novel aspects of TexTailor? The current version of TexTailor appears to be a combination of existing methods. It would be helpful if they could elaborate on any unique modifications within the resampling scheme or improvements made to accelerate the fine-tuning process.
2. While the paper discusses performance preservation loss qualitatively, a quantitative analysis of its impact on quality would clarify its specific role. Including an ablation study of the performance preservation loss in Table 2 could better highlight its contribution to TexTailor’s performance.
3. Since the viewpoint refinement uses a fixed threshold, how sensitive is the model’s performance to changes in this parameter?

---

> ### Author Response · Authors · 2024-11-24
>
> Thank you for taking the time to review our work. We greatly appreciate your recognition of the key aspects of our proposed methods and your positive feedback on the experiments. We will revise the manuscript to incorporate your suggestions.
>
> >Q1. While effective, the approach primarily combines existing techniques, with limited emphasis on novel contributions. The paper could be strengthened by enhancing the resampling scheme or accelerating the fine-tuning phase.
>
> I believe the contributions of this paper are as follows:
>
> 1.	We extend the resampling technique, previously used only in 2D image inpainting, to the field of 3D texture synthesis by applying it to the DDIM non-Markovian process.
>
> 2.	Without relying on an external dataset of 3D meshes, textures, or text, we propose a novel approach that fine-tunes the model using only a few images that accurately represent the texture of a specific object within the distribution learned by the existing depth-aware T2I model. Even with just a few images inferred by the original model, this approach effectively compensates for textures that are difficult to generate at certain angles and demonstrates the ability to maintain texture consistency across various viewpoints
>
> 3.	To address the catastrophic forgetting phenomenon—where a depth-aware T2I model forgets information originally learned from a large-scale dataset when fine-tuned on a smaller one—we propose the performance preservation loss to mitigate this issue and maintain the model's performance.
>
> 4.	To eliminate the need for manually configuring optimal camera positions based on object geometry—a process requiring significant time and effort—we introduce an adaptive method that adjusts camera positions dynamically based on the extent of texture coverage at each viewpoint.
>
> As you mentioned, point 1 may seem less novel as it recombines existing methods. However, we believe that the other contributions, particularly the ability to generate consistent images across various angles, can be highly applicable not only to the texture synthesis field but also to other 3D domains requiring such consistency. This makes our work a valuable contribution to the ICLR community and a potential foundation for future research.
>
> We anticipate that combining LoRA with ControlNet could shorten the fine-tuning phase while maintaining quality. However, due to time and resource constraints during the rebuttal period, we are unable to conduct all the necessary experiments at this time. We plan to explore this approach as part of our future work.
>
> Additionally, to further demonstrate the effectiveness of our proposed methodology, we will showcase qualitative comparisons for four additional objects from the Objaverse dataset, beyond the four objects already presented in the qualitative results of the paper, in GIF format. Furthermore, we will include qualitative comparison results for clothed human meshes from RenderPeople [10], also presented in GIF format.
>
> The text prompts for RenderPeople's clothed human meshes were created based on the ground truth images provided by RenderPeople, which were designed by professional designers.

---

> ### Author Response · Authors · 2024-11-24
>
> #### 1.Objaverse Dataset
> |Text Prompt|Latent-Paint|Text2Tex|TEXTure|Paint-it|Ours|
> |:-----:|:-----:|:-----:|:-----:|:-----:|:-----:|
> |"a basketball"|[link](https://drive.google.com/file/d/1PUhh1CFVGfzvHqeNS7-VWlspWygV7M9i/view?usp=drive_link)|[link](https://drive.google.com/file/d/10-Vjeq2lBFIrApNeJ409yoJL3tbdL7M5/view?usp=drive_link)|[link](https://drive.google.com/file/d/1-zmxggvpEi17NlXk4rwqswcguWHToUnx/view?usp=drive_link)|[link](https://drive.google.com/file/d/1tGnNDX-1Z1AWRACQzloKAnW7EJrbkIgV/view?usp=drive_link)|[link](https://drive.google.com/file/d/1ChOOEJCHugwlXSTZMtRPVyR7MOBsXiow/view?usp=drive_link)|
> |"a cd player"|[link](https://drive.google.com/file/d/1fKhj-8aeGkVKjfb3FwtGUBkuf7FuKiM2/view?usp=sharing)|[link](https://drive.google.com/file/d/1VdXOQslsVru-x2x19tGYfKttT9rI7hit/view?usp=sharing)|[link](https://drive.google.com/file/d/1JE_LDcZm1-oBX4QJN-dIlmBCcNzJz2ne/view?usp=sharing)|[link](https://drive.google.com/file/d/1sRkv6iXBa7e9TVSMcV3hrlbZ8tZo0u_m/view?usp=sharing)|[link](https://drive.google.com/file/d/17k4XOBHya95ZsYwqRs_NONUff9Apw7r4/view?usp=sharing)|
> |"a desk chair"|[link](https://drive.google.com/file/d/1mm_0cvW2RsvktsLcy07JXAeS5CGVfENO/view?usp=drive_link)|[link](https://drive.google.com/file/d/1yJQkmI6THlNn3hoL-mW2O7D5CI_A6gq9/view?usp=drive_link)|[link](https://drive.google.com/file/d/1pRvTW1eEjVNk7rycoJRAJ8IdvPqHEj46/view?usp=drive_link)|[link](https://drive.google.com/file/d/1Jpx9vtBIV7ymD7zRHLBIPSNAxSItMjFe/view?usp=drive_link)|[link](https://drive.google.com/file/d/1ELVqtUhQbT5XmUcqko0h1pJjMGaicmAl/view?usp=drive_link)|
> |"a dumpster"|[link](https://drive.google.com/file/d/1HKaoGX_dUN0-TQXlmIAuCbHygUFUV3bD/view?usp=drive_link)|[link](https://drive.google.com/file/d/1f1m09_mCKa9_4KZOGKme8BwsAc_7lkJG/view?usp=drive_link)|[link](https://drive.google.com/file/d/18UZWI-sGhUqXNuFW0SdnIFFO_QNVoiNI/view?usp=drive_link)|[link](https://drive.google.com/file/d/1zN_VombixmEsueJfd6L5kP7zws5fzq0G/view?usp=drive_link)|[link](https://drive.google.com/file/d/1jtiXfphExSoW4wGiDGUBRTD1t4nGVdKi/view?usp=drive_link)|
> |"a hammer"|[link](https://drive.google.com/file/d/1RSq-vzcPfclttJToHObHypn7hs17VLEd/view?usp=sharing)|[link](https://drive.google.com/file/d/1Kw7aEhSdx3mCXdYkGKRzrGeobqaLJwAW/view?usp=sharing)|[link](https://drive.google.com/file/d/1aexhUd7Nl0Aey3Vg4I-XnoAdlzb3UZPJ/view?usp=drive_link)|[link](https://drive.google.com/file/d/1kxIuyKW3ROjT25KaohiRxpwVAwvQaVWC/view?usp=sharing)|[link](https://drive.google.com/file/d/1GqPePJEubcHmS5LOnMir5OmVIsHv8iTL/view?usp=drive_link)|
> |"a minivan"|[link](https://drive.google.com/file/d/1yEG9Ni4rrNaZ4kCvL4WRgrY5nU2CxfZS/view?usp=drive_link)|[link](https://drive.google.com/file/d/11nAvcQqOKF-fCaP5-ZAHBNTzD27Fujz2/view?usp=drive_link)|[link](https://drive.google.com/file/d/1sUQbNuGWEw9S85RozmnzII0ByIneFv3W/view?usp=drive_link)|[link](https://drive.google.com/file/d/1ku15fjR4ts_2KJm0Rl6OqGi5h9mP3hqi/view?usp=drive_link)|[link](https://drive.google.com/file/d/1EJcNOro3_wlw6spZ7TYFwtWfZ1Vx3dbL/view?usp=drive_link)|
> |"a money"|[front](https://drive.google.com/file/d/10rPPgZsWc0iotldevklujifOZMbiz-0X/view?usp=drive_link), [back](https://drive.google.com/file/d/1PaRrWMzpS073RU7bg8NKvnB2wVvnF1ZX/view?usp=drive_link)|[front](https://drive.google.com/file/d/1uNwXkh7O4UASFmm_BhDJJfX8MYWj_My6/view?usp=drive_link), [back](https://drive.google.com/file/d/10x-gy15J7OgJdymqFsalM_R-aF-XCJi4/view?usp=drive_link)|[front](https://drive.google.com/file/d/1rp4m1AdduMMOUyXMHUng5eq8jmQBsF1e/view?usp=drive_link), [back](https://drive.google.com/file/d/1mRWQlFv7YF4SeOrPSq4GrXkro6nveYzC/view?usp=drive_link)|[front](https://drive.google.com/file/d/1EFDwk-csQEiokhU09NiubX-yGC4NNXr7/view?usp=drive_link), [back](https://drive.google.com/file/d/1I_uEbvFRMn9HdfhjgDSNf7miW_fgTLQf/view?usp=drive_link)|[front](https://drive.google.com/file/d/1SRVIlpRovqU2o2d1XYQMkdHXsgd_TQxC/view?usp=drive_link), [back](https://drive.google.com/file/d/1Oc45Uj5M5ySbKikOYqvl6jA0KX9sEB6O/view?usp=drive_link)|
> |"a sushi"|[link](https://drive.google.com/file/d/15QWzVwVvJ-r7rFqUcVXGhnmNGRMdUClx/view?usp=drive_link)|[link](https://drive.google.com/file/d/1d7hxtTi8NFio4kAXeMJjIGyrIaOwmfmA/view?usp=drive_link)|[link](https://drive.google.com/file/d/1uZerJAnPSwV-bAw0e3JXwBpaPsL96XzK/view?usp=drive_link)|[link](https://drive.google.com/file/d/1VMeuBZoB-CuDDc0ekSCb5ezrXQLeVNgr/view?usp=drive_link)|[link](https://drive.google.com/file/d/1gSCl1HYbhxIK4ovEEs0OJVF1CgqIgHLg/view?usp=drive_link)|

---

> ### Author Response · Authors · 2024-11-24
>
> #### 2.RenderPeople Dataset
> |Text Prompt|Latent-Paint|Text2Tex|TEXTure|Paint-it|Ours|
> |:-----:|:-----:|:-----:|:-----:|:-----:|:-----:|
> |"A business woman wearing a white blouse with a ribbon detail, light beige pants, nude-tone heels, and neatly tied blonde hair"|[claudia](https://drive.google.com/file/d/1cugJNiHxkIxzZmfapOkUSUt1b5nmX-I4/view?usp=drive_link)|[claudia](https://drive.google.com/file/d/1tuUSCLRtZs8meKal50_kkfX0XkTDIlwU/view?usp=drive_link)|[claudia](https://drive.google.com/file/d/1HkO7HNKWkFh4u_xZfxRTgiyQV35ENyvS/view?usp=drive_link)|[claudia](https://drive.google.com/file/d/1cYYHgVoRn5YN-BQGbSXxtVa3ONWLBpIU/view?usp=drive_link)|[claudia](https://drive.google.com/file/d/1jpUILlxb39eSsAhZnk0kMBt2dTQ_iswl/view?usp=drive_link)|
> |"A man, wearing a white dress shirt, a black vest, black formal trousers, a black tie, a black belt, and dark formal shoes, with short neatly styled hair"|[eric](https://drive.google.com/file/d/1zu8JFHMlV2oHCu84je9z-hJn1_YUACl0/view?usp=drive_link)|[eric](https://drive.google.com/file/d/1ejcq1bB6djVUmmCqAxbqrDA2q_l_mOsK/view?usp=drive_link)|[eric](https://drive.google.com/file/d/1kQ4JW-Nu14p8qiLvR4RwvB90Iu_elRoI/view?usp=drive_link)|[eric](https://drive.google.com/file/d/1FItS8NhnffnzjoNPPU1or_t6ScWUveO2/view?usp=drive_link)|[eric](https://drive.google.com/file/d/1YibXGY7AicDrCaalaJUChKGRE5nQ6T7c/view?usp=drive_link)|
> |"A man, wearing a gray short-sleeve T-shirt, blue jeans, white sneakers, and short, dark brown hair styled neatly"|[manuel](https://drive.google.com/file/d/1yRmwmkkZ8c2VmC3fr2B92R2ztWT6BC0a/view?usp=sharing)|[manuel](https://drive.google.com/file/d/1jasYCugvfEaWtXMusmCFExluKTggHtKX/view?usp=sharing)|[manuel](https://drive.google.com/file/d/1kXzUleXzPhJkehtuoNUPhcj9YJzqYSRm/view?usp=sharing)|[manuel](https://drive.google.com/file/d/1ZYkWlhq2cxa-jyxaPeHbdvqTmjQE9-RP/view?usp=sharing)|[manuel](https://drive.google.com/file/d/1epeP2Ms2HC5AkW7uumw9f08H6dk-FVCc/view?usp=sharing)|
> |"A woman with medium-dark skin tone, wearing a black blazer, a black top, gray pants with a gray tied belt, black heels, and having neatly styled dark hair"|[carla](https://drive.google.com/file/d/1SN1Hft8lAbLU4HHTxB7k6e_WxoG-h-r-/view?usp=drive_link)|[carla](https://drive.google.com/file/d/1jJCNNpgXsB8xJKu6NnGkXicQ3ZDUOCRf/view?usp=drive_link)|[carla](https://drive.google.com/file/d/1OZ-uWrrX06M19CDgpukyCMkM0A5A4l78/view?usp=drive_link)|[carla](https://drive.google.com/file/d/1vc3EaAvNO18t8bUo8l_-KC5hhGTOzDTD/view?usp=sharing)|[carla](https://drive.google.com/file/d/1TlZ2wcokdUOz9Q-KIsSIShCiheelB1ox/view?usp=sharing)|

---

> ### Author Response · Authors · 2024-11-25
>
> > Q2. While the paper discusses performance preservation loss qualitatively, a quantitative analysis of its impact on quality would clarify its specific role. Including an ablation study of the performance preservation loss in Table 2 could better highlight its contribution to TexTailor’s performance.
>
> Here is the table presenting the ablation study results for the performance preservation loss, as per your suggestion.
>
> |w/Resampling|w/Training|w/Performance preservation loss|w/Adaptive view refinement|LPIPS $\downarrow$|FID $\downarrow$|
> |:-----:|:-----:|:-----:|:-----:|:-----:|:-----:|
> |$\checkmark$|x|x|x|38.89|30.924|
> |$\checkmark$|$\checkmark$|x|x|39.849|53.85|
> |$\checkmark$|$\checkmark$|$\checkmark$|x|38.00|30.567|
> |$\checkmark$|$\checkmark$|$\checkmark$|$\checkmark$|37.89|29.998|
>
> As shown in the results, fine-tuning without introducing the performance preservation loss causes the existing capabilities of the depth-aware diffusion model to deteriorate, leading to worse performance metrics. These findings are also supported qualitatively, further reinforcing our claims.
>
> > Q3. Since the viewpoint refinement uses a fixed threshold, how sensitive is the model’s performance to changes in this parameter?
>
> Thank you for raising this insightful question. We agree that evaluating the sensitivity of the model's performance to changes in the fixed threshold used in viewpoint refinement would provide valuable insights. However, due to time and resource constraints during the rebuttal period, we were unable to conduct these additional experiments. We acknowledge the importance of this analysis and plan to address it as part of our future work to further refine and validate our approach. We sincerely appreciate your thoughtful feedback.

---

> > ### Comment · Reviewer_uECh · 2024-11-26
> >
> > Thank you for the detailed response to my review and the concerns raised by other reviewers. While I still find the contribution somewhat limited, most of my initial concerns have been addressed, and I will adjust my rating upward accordingly.

---

### Author Response · Authors · 2024-11-27

We deeply appreciate the time and effort each reviewer has devoted to providing thoughtful and constructive feedback. We have carefully addressed the reviewers’ comments and submitted a revised version of the paper that incorporates their valuable suggestions.

To summarize the changes made:

(1) Improved the flow of the introductory section on page 1, line 46.

(2) Replaced parts of Figure 1, Figure 6, and Figure 7 to address quality degradation caused by scaling.

(3) Added quantitative results for the performance preservation loss ablation study in Table 2.

(4) Removed statements claiming our results are better aligned with prompts in the qualitative comparisons section.

(5) Expanded the contribution section for greater clarity.

(6) Added citations for the methods compared in Figure 1 to the caption and clarified the meaning of the equation in Figure 3.

(7) Included additional qualitative comparisons for Objaverse and RenderPeople datasets in the appendix.

(8) Added qualitative results for non-diffuse objects in the appendix.

(9) Provided ablation studies on various objects in the appendix.

(10) Added a detailed discussion of limitations in the appendix.

(11) Included zoomed-in comparisons between TexTailor and Text2Tex in the appendix.

(12) Corrected minor typographical errors throughout the manuscript.

(13)  included GIF results in the supplementary material

We sincerely thank the reviewers for their invaluable insights and guidance, which have significantly improved the quality of our work.

---

### Meta-Review · Area_Chair_t5Dm · 2024-12-21

**Metareview:**

Summary:
- This paper presents a method for creating view-consistent texture on a 3D object using text prompts. The basic approach follows an inpainting framework and proposes a resampling method and fine-tuning to improve texture consistency across viewpoints.

Strength:
- "solid and correct, analysis is thorough and well-motivated"
- well-written paper.

Weakness:
- This paper primarily combines existing techniques. The technical contributions are limited.

Justification:
- All four reviewers are leaning positive about this paper, particularly after the rebuttal, where the authors' rebuttal adequately addressed several initial concerns from the reviewers. While the paper does not present intrinsically novel techniques, the integration of existing methods is well-motivated and well-executed. The evaluation is solid and validates the claims of improved view-consistent texture generation. The AC agrees with the reviewers and recommends to accept.

**Additional Comments On Reviewer Discussion:**

Reviewer uECh asked for classifications of the paper contributions, ablation study for performance preservation loss, and the sensitivity in the threshold parameter.

The authors provided a detailed response with additional ablation study results. Reviewer uECh still considered the paper's technical contribution incremental but is satisfied with the responses. The rebuttal did not address the sensitivity in the threshold parameter, but the AC considers a minor issue.

Reviewer 3JDj commented on the novelty of the resampling scheme, insufficient results (e.g., 3D human texturing), and the metrics. The rebuttal responded by providing additional results on the RenderPeople dataset, additional types of objects, and runtime/memory comparisons. Reviewer 3JDj still has concerns about the scope and contribution but decided to raise the rating since most initial concerns were adequately addressed.

Reviewer ujQC was initially negative about this work, highlighting issues in the method exposition and experiments. Reviewer ujQC finds the explanations about the metrics convincing and raises the rating to borderline positive.

Reviewer NBoV comments that the resampling strategy is similar to several prior work and asks for clarification. Reviewer NBoV did not respond to the author's rebuttal.

Overall, the reviewers appreciate the authors' detailed responses on the initial concerns. The additional explanations and results further validate the method's effectiveness and improvement over prior art. The rebuttal successfully convinced two reviewers to increase their scores.

---

### Decision · Program_Chairs · 2025-01-22

Accept (Poster)